# Elucidating Human Milk Oligosaccharide biosynthetic genes through network-based multi-omics integration

Benjamin P. Kellman[1,2,3,7], Anne Richelle[1,7], Jeong-Yeh Yang[4], Digantkumar Chapla[4], Austin W. T. Chiang [1], Julia A. Najera[1], Chenguang Liang[1,3], Annalee Fürst[1], Bokan Bao [1,2,3], Natalia Koga[1], Mahmoud A. Mohammad [5], Anders Bech Bruntse[1], Morey W. Haymond[5], Kelley W. Moremen [4], Lars Bode[1,6] & Nathan E. Lewis [1,3✉]

Human Milk Oligosaccharides (HMOs) are abundant carbohydrates fundamental to infant health and development. Although these oligosaccharides were discovered more than half a century ago, their biosynthesis in the mammary gland remains largely uncharacterized. Here, we use a systems biology framework that integrates glycan and RNA expression data to construct an HMO biosynthetic network and predict glycosyltransferases involved. To accomplish this, we construct models describing the most likely pathways for the synthesis of the oligosaccharides accounting for >95% of the HMO content in human milk. Through our models, we propose candidate genes for elongation, branching, fucosylation, and sialylation of HMOs. Our model aggregation approach recovers 2 of 2 previously known gene-enzyme relations and 2 of 3 empirically confirmed gene-enzyme relations. The top genes we propose for the remaining 5 linkage reactions are consistent with previously published literature. These results provide the molecular basis of HMO biosynthesis necessary to guide progress in HMO research and application with the goal of understanding and improving infant health and development.

---

[1] Department of Pediatrics, University of California, San Diego, La Jolla, CA 92093, USA. [2] Bioinformatics and Systems Biology Graduate Program, University of California, San Diego, La Jolla, CA 92093, USA. [3] Department of Bioengineering, University of California, San Diego, La Jolla, CA 92093, USA. [4] Complex Carbohydrate Research Center, University of Georgia, Athens, GA, USA. [5] Department of Pediatrics, Children's Nutrition Research Center, US Department of Agriculture/Agricultural Research Service, Baylor College of Medicine, Houston, TX 77030, USA. [6] Larsson-Rosenquist Foundation Mother-Milk-Infant Center of Research Excellence (MOMI CORE), University of California, San Diego, La Jolla, CA 92093, USA. [7] These authors contributed equally: Benjamin P. Kellman, Anne Richelle. ✉email: nlewisres@ucsd.edu

Human milk is the "gold standard" of nutrition during early life[1–3]. Beyond lactose, lipids, and proteins, human milk contains 11–17% (dry weight) oligosaccharides (Human Milk Oligosaccharides, HMOs)[4,5]. HMOs are milk bioactives known to improve infant immediate and long-term health and development[6]. HMOs are metabolic substrates for specific beneficial bacteria (e.g., *Lactobacillus* spp. and *Bifidobacter* spp.), and shape the infant's gut microbiome[2,7]. HMOs also impact the infant's immune system, protect the infant from intestinal and immunological disorders (e.g., necrotizing enterocolitis, HIV, etc.), and may aid in proper brain development and cognition[2,6,8,9]. In addition, recent discoveries show that some HMOs can be beneficial to humans of all ages, e.g. the HMO 2'-fucosyllactose (2'FL) protecting against alcohol-induced liver disease[10].

The biological functions of HMOs are determined by their structures[6]. HMOs are unconjugated glycans consisting of 3–20 total monosaccharides drawn from 3–5 unique monosaccharides: galactose (Gal, A), glucose (Glc, G), N-acetylglucosamine (GlcNAc, GN), fucose (Fuc, F), and the sialic acid N-acetyl-neuraminic acid (NeuAc, NN) (Fig. 1A). All HMOs extend from a common lactose (Galβ1-4Glc) core. The core lactose can be extended at the non-reducing end, with a β-1,3-GlcNAc to form a trisaccharide. That intermediate trisaccharide is quickly extended on its non-reducing terminus with a β-1,3-linked galactose to form a type-I tetrasaccharide (LNT) or a β-1,4-linked galactose to form a type-II tetrasaccharide (LNnT). Additional branching of the trisaccharide or tetrasaccharide typically occurs at the lactose core by addition of a β-1,6-linked GlcNAc to the Gal residue. Lactose or the elongated oligosaccharides can be further fucosylated in an α-1,2-linkage to the terminal Gal residue, or α-1,3/4-fucosylated on internal Glc or GlcNAc residues, and α-2,3-sialylated on the terminal Gal residue or α-2,6-sialylated on external Gal or internal GlcNAc residues[6,8] (Fig. 1B).

Despite decades of study, many details of HMO biosynthesis remain unclear. While the many possible monosaccharide addition events above are known, the order of the biosynthetic steps and many of the enzymes involved are unclear (Table 1). For example, the lactose core is extended by alternating actions of β-1,3-N-acetylglucosaminyltransferases (b3GnT) and β-1,4-galactosaminyl-transferases (b4GalT) while β-galactoside sialyltransferases (SGalT) and α-1,2-fucosyltransferases (including the FUT2 'secretor' locus) are responsible for some sialylation and fucosylation of a terminal galactose, respectively[11]. However, each enzymatic activity in HMO extension and branching can potentially be catalyzed by multiple isozymes in the respective gene family. Direct evidence of the specific isozymes performing each reaction in vivo is limited.

Here we leverage the heterogeneity in HMO composition and gene expression across human subjects to refine our knowledge of the HMO biosynthetic network. Milk samples were collected from 11 lactating women across two independent cohorts between the 1st and 42nd day post-partum (see Methods). Gene expression profiling of mammary epithelial cells was obtained from mRNA present in the milk fat globule membrane interspace (Supplementary Dataset 1 and Supplementary Figs. 1, 2). Absolute (using commercial standards) and relative (normalized to total HMO weight in a sample) concentrations of the 16 most abundant HMOs were measured; these 16 HMOs typically account for >95% of HMO mass in a milk sample[12] (Supplementary Dataset 1 and Supplementary Fig. 3). Starting from a scaffold of all possible reactions[13–18], we used constraint-based modeling[19,20] to reduce the network to a set of relevant reactions and most plausible HMO structures when not known[21] to form the basis for a mechanistic model. This resulted in a ranked ensemble of candidate biosynthetic pathway topologies. We then ranked 44 million candidate biosynthesis networks to identify the most likely network topologies and candidate enzymes for each reaction by integrating sample-matched transcriptomic and glycoprofiling data from the 11 subjects. For this we simulated all reaction fluxes and tested the consistency between changes in flux and gene expression to determine the most probable gene isoform

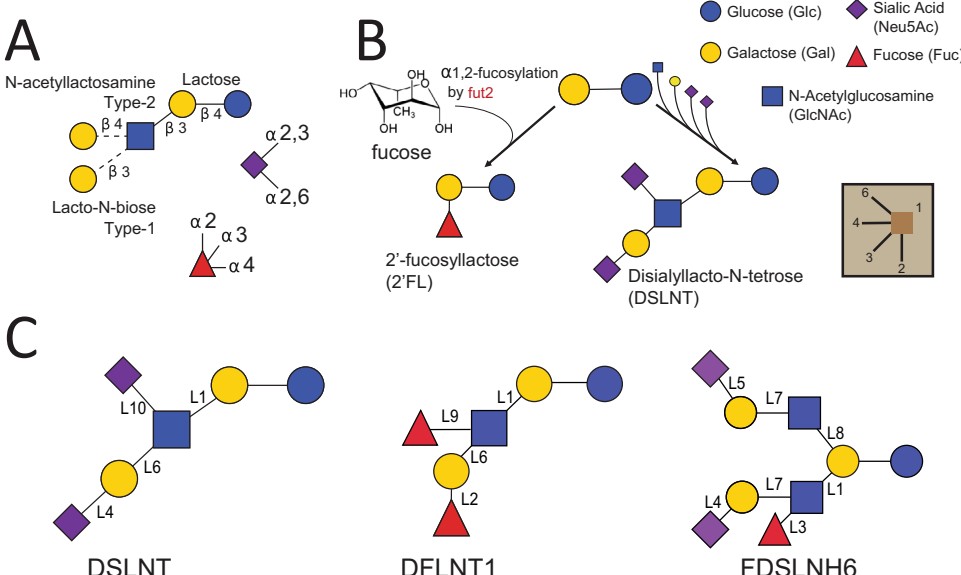

**Fig. 1 HMO blueprint and synthesis. A** HMOs are built from a combination of the five monosaccharides D-glucose (Glc, blue circle), D-galactose (Gal, yellow circle), N-acetyl-glucosamine (GlcNac, blue square), L-fucose (Fuc, red triangle), and sialic acid (N-acetyl-neuraminic acid (NeuAc), purple diamond). Lactose (Gal-β-1,4-Glc) forms the reducing end and can be elongated with several Lacto-N-biose or N-acetyllactosamine repeat units (Gal-β-1,3/4-GlcNAc). Lactose or the polylactosamine backbone can be fucosylated with α-1,2-, α-1,3-, or α-1,4- linkages or sialylated in α-2,3- or α-2,6- linkages[2]. **B** Small HMOs can be fucosylated to make 2'FL while larger HMOs can be synthesized by the extension of the core lactose with N-acetyllactosamine (type-I) or lacto-N-biose (type-II) and subsequent decoration of the extended core with sialic acid to make more complex HMOs, such as DSLNT. **C** Three HMOs in this study: DSLNT, isomer 1 of DFLNT, isomer 6 of FDSLNH; isomer structures represent predictions from this study (see Methods, Supplementary Fig. 5). Each monosaccharide-linking glycosidic bond is labeled (L1, L2,...L10) according to the linkage reactions listed in Table 1.

**Table 1 Glycosylation reactions examined.**

| Linkage | Reaction | EC Identifier | Acceptor {Constraint} | Product | Candidates |
|---|---|---|---|---|---|
| L1:b3GnT | b-1,3 N-acetylglucosamine | 2.4.1.149 | (A | (GNb3A | B3GNT2-6,8-9 |
| L2:a2FucT | a-1,2 fucosyltransferase | (2.4.1.69,344) | (A | (Fa2A | FUT1-2 |
| L3:a3FucT | a-1,3 fucosyltransferase | (2.4.1.152) | G/GN {~Ab3GN} | Fa3G/GN | FUT3-7,9-11 |
| L4:ST3GalT | (b-Gal) a-2,3 sialytransferase | (2.4.99.4) | (A | (NNa3A | ST3GAL1-6 |
| L5:ST6GalT | (b-Gal) a-2,6 sialytransferase | 2.4.99.1 | (A | (NNa6A | ST6GAL1-2 |
| L6:b3GalT | b-1,3 galactotransferase | 2.4.1.86 | (GN | (Ab3GN | B3GALT1-2,4-5 |
| L7:b4GalT | b-1,4 galactotransferase | 2.4.1.90 | (GN | (Ab4GN | B4GALT1-6 |
| L8:b6GnT | b-1,6 N-acetylglucosamine | (2.4.1.150) | GNb3Ab4G | GNb3(GNb6)Ab4G | GCNT1-4,7 |
| L9:a4FucT | a-1,4 fucosyltransferase | 2.4.1.65 | Ab3GNb3A {~GNb4Ab3GNb3A} | Ab3(Fa4)GNb3A | FUT3,5 |
| L10:ST6GnT | (b-1,3-GlcNac) a-2,6 sialytransferase | (2.4.99.3,7) | Ab3GNb3A | Ab3(NNa6)GNb3A | ST6GALNAC1-6 |

We studied several candidate glycosyltransferases expressed in cohort 1 and 2 (described in the next section) to identify candidates for 10 elementary reactions (see Methods section, Supplementary Dataset 2). Acceptor, product and constraint are represented in LiCoRR[11]: monosaccharides include Gal (A), Fuc (F), Glc (G), GlcNAc (GN), Neu5Ac (NN). Additionally, ")" and "(" indicate initiation and termination of a branch respectively, "[X/Y]" indicates either monosaccharide, and "~" indicates a negation. An asterisk "*" indicates an imperfect match between the EC number and reaction. We note that gene "candidates" for each reaction (last column) were not used to inform the biosynthetic model construction. Candidate genes are those compared to completed biosynthetic models to evaluate consistency between candidate gene expression and simulated flux through the corresponding reaction.

responsible for each linkage type. We followed with direct observations through fluorescence activity assays to confirm our predictions. Finally, we performed transcription factor analysis to delineate regulators of the system. The resulting knowledge of the biosynthetic network can guide efforts to unravel the genetic basis of variations in HMO composition across subjects, populations, and disorders using systems biology modeling techniques.

## Results

**HMO abundances do not correlate with known enzyme expression.** While α-1,2-fucosylation of glycans in humans can be accomplished by both FUT1 and FUT2, only FUT2 is expressed in mammary gland epithelial cells (Supplementary Dataset 2). FUT2, the "secretor" gene, is essential to ABH antigens[22–24] as well as HMO[2,25,26] expression. We confirmed that non-functional FUT2 in "non-secretor" subjects guarantees the near-absence of α-1,2-fucosylated HMOs like 2′FL and LNFP1 (Fig. 2C). However, examining only subjects with functional FUT2 (Secretors), we found FUT2 expression levels and the concentration (nmol/ml) of HMOs containing α-1,2-fucosylation do not correlate in sample-matched microarray (Supplementary Figs. 1, 2) and HMO abundance measurements by HPLC (Fig. 2, Supplementary Fig. 3, and Supplementary Dataset 1). Generalized Estimating Equations (GEE) showed no significant positive association (2′FL Wald $p = 0.056$; LNFPI Wald $p = 0.34$). FUT1 could catalyze this reaction but its expression was not detected in these samples. We hypothesized that to successfully connect gene expression to HMO synthesis, one must account for all biosynthetic steps and not solely rely on direct correlations.

**High-performing candidate biosynthetic models are supported by gene expression and predicted model flux across subjects.** To determine which candidate genes (Supplementary Fig. 4) support HMO biosynthesis, we built and examined models for HMO biosynthesis in human mammary gland epithelial cells (See Supplementary Methods 4.1–4.3 and 5.4 for complete details). From the basic reaction set (Fig. 3A), we generated the complete reaction network (Fig. 3B) containing all possible reactions and HMOs with up to nine monosaccharides (Supplementary Fig. 5). The Complete Network was trimmed to obtain a Reduced Network (Fig. 3D and Supplementary Dataset 3) by removing reactions unnecessary for producing the observed oligosaccharides. Candidate models (Fig. 3E and Supplementary Fig. 6) were built,

capable of uniquely recapitulating the glycoprofiling data from milk using two independent cohorts – cohort 1 with 8 samples from 6 mothers between 6 h and 42 days postpartum[27,28] and cohort 2 with 2 samples per mother on the 1st and second day after birth[29]. Mixed Integer Linear Programming (MILP) was used to identify subnetworks with the minimal number of reactions from the Reduced Network. We identified 44,984,988 candidate models that can synthesize the measured oligosaccharides. Each candidate model contains 43–54 reactions (19.5–24.4% of the reactions in the Reduced Network (Supplementary Table 1)). These models covered all the feasible combinations of HMO synthesis by the 10 known glycosyltransferase families (Fig. 1D) that could describe the synthesis of the HMOs in this study.

To identify the most likely biosynthetic pathways for HMOs, we computed a model score for each candidate model using the glycoprofiling and transcriptomic data from the two independent cohorts (Supplementary Figs. 7, 8), after excluding low-expression gene candidates. Genes were excluded when expression was undetected in over 75% of microarray samples and the independent RNA-Seq[30] measured low expression relative to the GTEx[31]: TPM < 2 and 75th percentile Lemay <GTEx Median TPM. Specificity and expression filtration reduced the candidate genes from 54 to 24 (see Supplementary Results, Supplementary Dataset 2, Supplementary Fig. 4); three linkages (L2, L5, and L9) were resolved by filtration alone indicating that FUT2, ST6GAL1, and FUT3 respectively perform these reactions.

Following low-expression filtering, we compared flux-expression correlation. Leveraging sample-matched transcriptomics and glycomics datasets, we computed model scores indicating the capacity of each candidate gene to support corresponding reaction flux. The model score was computed by first identifying for each reaction, the candidate gene that shows the best Spearman correlation between gene expression and normalized flux; flux was normalized as a fraction of the input flux to limit the influence of upstream reactions (Supplementary Figs. 7, 8 and Supplementary Methods 4.4). The highest gene-linkage scores, for each reaction, for each model were averaged to obtain a model score (Fig. 3G, see Methods section). The model scores indicate consistency between gene expression and model-predicted flux. The high-performing models (z(model score)>1.646) were selected for further examination (Fig. 3H, see Methods section). Though quantile-quantile plots indicated the model score distributions were pseudo-Gaussian, variation in skew resulted in slightly different numbers of high-performing models for the two different subject

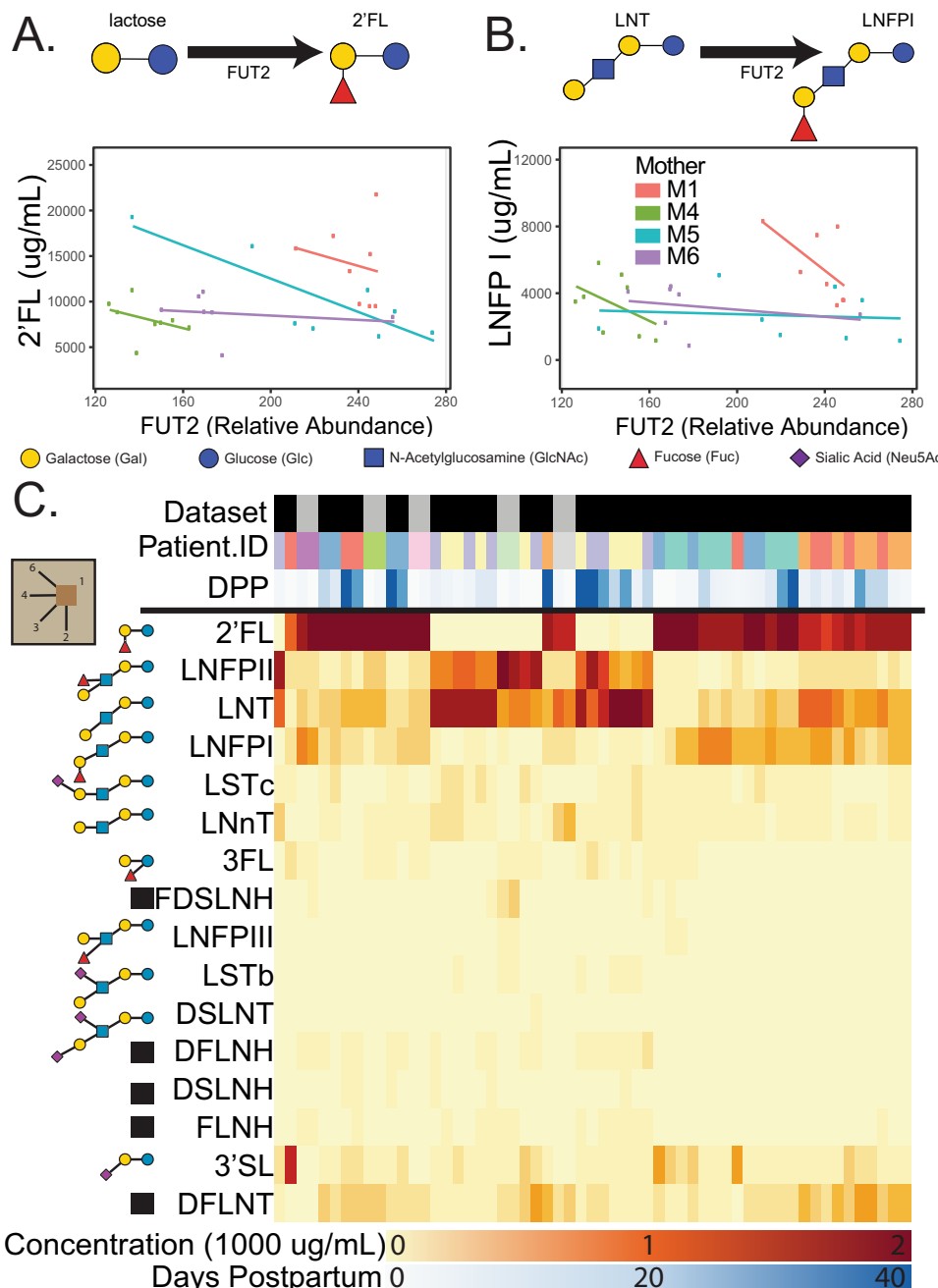

**Fig. 2 FUT2 expression should increase 2′FL and LNFPI which require the enzyme but there is no significant positive association. A concentration heatmap for all HMOs is shown below.** Direct comparison of FUT2 gene expression and concentrations (nmol/mL) of α-1,2-fucose containing HMOs, 2′FL (**A**) and LNFPI (**B**), in sample-matched microarray and absolute HMO abundances reveal no significant association in secretor women from cohort 1 sampled between day 1 and 42 postpartum. The linear trends were drawn to illustrate poor HMO-gene correlations. Trendlines and points are colored by subject. Linear trends were used to illustrate the intuition of the GEE approach used to estimate these associations across subjects. Non-secretor mothers were excluded due to non-functional FUT2. **C** A heatmap of all HMO concentrations across cohort 1 and cohort 2 (top-bar black and gray respectively). Known HMO structures[2] are shown to the left of each row while undetermined structures are indicated with a black box. For literature curated[6,8,33,104,108,109] isomers of undetermined structures, see Supplementary Fig. 5.

cohorts. Specifically, we found 2,658,052 high-performing models from cohort 1 and 2,322,262 high-performing models using cohort 2 (Fig. 3 and Supplementary Table 2). We found 241,589 high-performing models common to cohort 1 and cohort 2. The model scores of commonly high-performing models are significantly correlated (Spearman $R_s = 0.2$, $p < 2.2e-16$) and a hypergeometric enrichment of cohort 1 and cohort 2 selected models shows the overlap is significant relative to the background of 44 million models (hypergeometric enrichment $p < 2.2e-16$). We analyzed

these 241,589 commonly high-performing models and determined which candidate genes were common in high-performing models (Supplementary Fig. 8).

To determine the most important reactions (Figs. 4 and 9, 10) in the Reduced Network, we asked which reactions were most significantly and frequently represented among the top 241,589 high-performing models. We then filtered to retain only the top 5% of most important paths from lactose to each observed HMO (see Methods section). The most important reactions form the

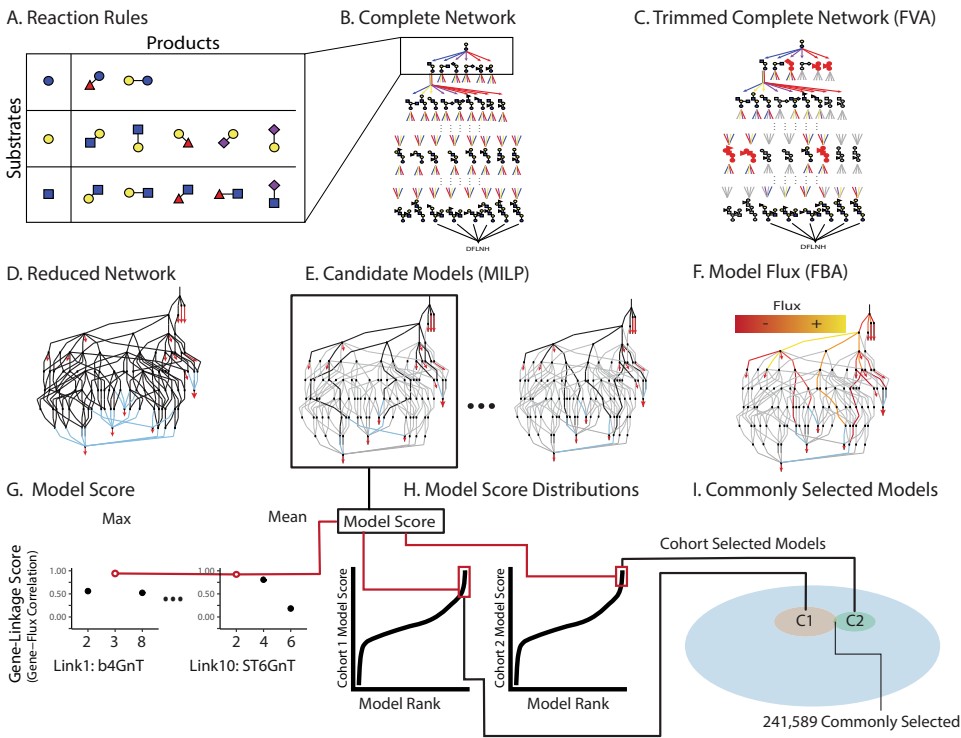

**Fig. 3 HMO biosynthesis models constructed with flux analysis and ranked by expression concordance perform consistently across cohorts. Overview of computational methods for model assembly (A–F) and assessment (G–I). A** To build the candidate models of HMO biosynthesis, reaction rules were defined to specify all possible monosaccharide additions. **B** The Complete Network includes all oligosaccharides and irreversible reactions resulting from the iterative addition of monosaccharides to a root lactose. **C** Using Flux Variability Analysis (see Supplementary Methods 5.4), the Complete Network was trimmed, removing reactions that cannot reach experimentally-measured HMOs, to produce a Reduced Network **D** (Supplementary Fig. 10); red triangles are observed HMOs blue lines are "sink reactions" joining alternative isomers (Supplementary Fig. 5). **E** From the Reduced Network, Mixed Integer Linear Programming (MILP) was used to extract candidate models, each representing a subnetwork capable of uniquely synthesizing the observed oligosaccharide profile using a minimal number of reactions; black lines are reactions retained in a candidate model. **F** Flux Balance Analysis estimated flux through each reaction necessary to simulate the measured relative oligosaccharide abundance (Supplementary methods 5.4). **G** Model scores were computed as the average maximum correlation between linkage-specific candidate genes and normalized flux through that linkage (Supplementary Fig. 7 and Supplementary Methods 4.4). **H** Model scores were parameterized on cohort 1 (left) and cohort 2 (right) data (see Methods section). High-performing models, 95th percentile of scores, are highlighted in red. **I** Of the >40 million models considered (blue), 2.66 and 2.32 million models were high-performing when parameterized on data from cohort 1 or cohort 2, respectively. Nearly 250,000 models consistently explained the relationship between predicted flux and expression data from both cohort 1 and cohort 2. These commonly selected models were analyzed for common structural features.

summary network (Fig. 4). Here, HMO biosynthesis naturally segregates into type-I backbone structures, with β-1,3-galactose addition to the GlcNAc-extended core lactose, and type-II structures, with β-1,4-galactose addition to the GlcNAc-extended core lactose. As expected, LNFPI, LNFPII, LSTb, and DSLNT segregate to the type-I pathway while LNFPIII and LSTc are found in the type-II pathway (see Methods section for HMO definitions). The summary network suggests resolutions to large structurally ambiguous HMOs (FLNH5, DFLNT2, DFLNH7, and DSLNH2) by highlighting their popularity in high-performing models. The summary network also shows three reactions of high comparable strength projecting from GlcNAc-β-1,3-lactose to LNT, LNnT, and a bi-GlcNAc-ylated lactose (HMO8, Fig. 4, and Supplementary Dataset 3) suggesting LNT may be bypassed through an early β-1,3-GlcNAc branching event; a previously postulated alternative path[32]. We checked for consistency with previous work[33] and found that (1) the single fucose on the reducing-end Glc residue is always α-1,3 linked, (2) for monofucosylated structures, the non-reducing terminal β-1,3-galactose is α-1,2-fucosylated, (3) all galactose on the β-1,6-GlcNAc is always β-1,4 linked while all galactose on the β-1,3-GlcNAc are either β-1,3/4 linked. With the exception FDSLNH1, (4) no fucose is found at the reducing end of a branch, and (5) all α-1,2-fucose appear on a β-1,3-galactose and

not β-1,4-galactose in monofucosylated structures with more than four monosaccharides; suggesting that FDSLNH1 is an unlikely isomer. The summary network also suggests that most HMOs have type-I LacNAc backbones.

**Glycosyltransferases are resolved by ranking reaction consistency across several metrics**. We further analyzed the high-performing models to identify the glycosyltransferases responsible for each step in HMO biosynthesis (Table 1). As described (Supplementary Results 6.1), not all members of a gene family were examined in this analysis. Some genes were excluded due to their well-characterized irrelevance (e.g., FUT8) and others, like FUT1, were excluded due to low expression in lactating breast epithelium (see Supplementary Dataset 2, Supplementary Methods and Results for the detailed inclusion criteria). To determine the genes preferred for each reaction, we used three metrics to quantify the association between candidate gene expression and predicted flux. These were (1) proportion (PROP - the relative proportion of models best explained by a candidate gene, Supplementary Figs. 11–15), (2) gene linkage score (GLS - the average Spearman correlation between gene expression and flux), and (3) model score contribution (MSC - an estimate of the gene-influence indicated by the Pearson correlation between model

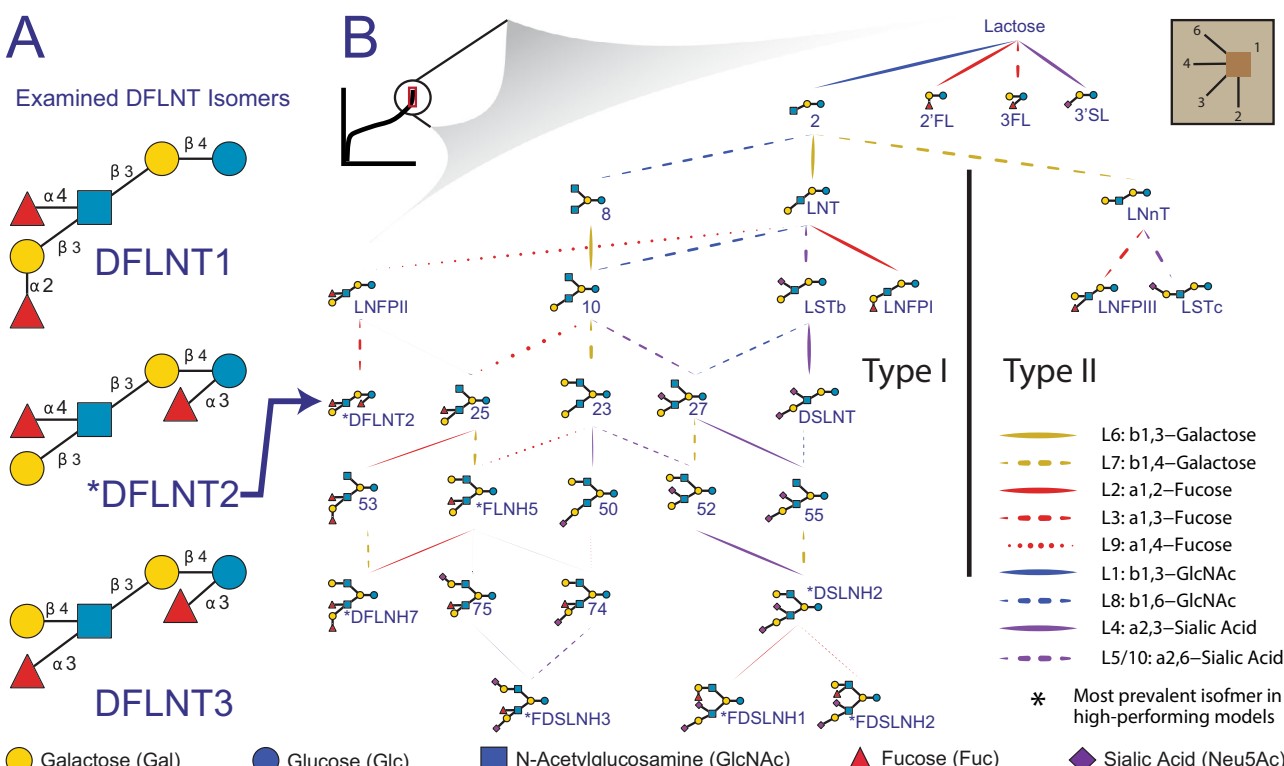

**Fig. 4 Summary network of the most important reactions in the Reduced Network.** Observed, intermediate and candidate HMOs most important to commonly high-performing networks were selected from the Reduced Network (Fig. 3D; Supplementary Methods 4.2). **A** Several ambiguous isomers (Supplementary Fig. 5) were preferred (Supplementary Fig. 14) in the commonly high-performing models. **B** A summary network was constructed from reaction importance; an aggregation of the proportion of high-performing models that include a reaction, and the enrichment of a reaction in the high-performing model set (see Methods section). Line weight indicates the relative importance of each reaction. Line color corresponds to the monosaccharide added at each step and line type corresponds to the linkage type. The summary network naturally segregates into type-I and type-II backbone structures. For measured HMO definitions (e.g., FDSLNH and DSLNT) see Methods section, for intermediate HMO definitions (e.g., 8, 10, or 25) see Supplementary Dataset 3, for uncertain structures (e.g. DFLNH7, FLNH5) see Supplementary Fig. 5.

score and gene linkage score) (Fig. 5A and Supplementary Fig. 12). For each candidate gene, we generated a reaction support score (Fig. 5B, see Methods section); the pooled significance of the maxima of PROP, GLS, and MSC across both cohorts.

Three reactions, L2 (FUT2), L5 (ST3GAL1), and L9 (FUT3), were matched to genes by default as they were the only gene candidates remaining following gene expression filtering (Supplementary Dataset 2, Supplementary Results). At least one gene showed significant support ($q < 0.1$) for each remaining reaction. GCNT3 shows highly significant support ($q < 0.001$) and nearly 100% of models selected this isoform over GCNT2C or GCNT1 (Supplementary Fig. 11). B4GALT4 is the most significantly supporting gene for the L7: b4GalT reaction (Fig. 5B). In both cohort 1 and 2, B4GALT4 outperforms all other isoforms in all three metrics. B4GALT4 expression best explains flux in 62 and 80% (PROP) of high-performing models using cohort 1 and 2 data respectively (Supplementary Fig. 11). B4GALT4 also has the highest MSC and GLS ($z > 5.6$) of any isoforms. Interestingly, while B4GALT1 is highly expressed and fundamental to lactose synthesis in the presence of α-lactalbumin and lactation in general[34,35], it showed negligible support for the L7 reaction (Fig. 5B). Considering the reaction support score, all linkages show at least one gene for each reaction that significantly explains behavior across cohorts (Fig. 5B).

**Kinetic assays corroborate gene-reaction associations.** Towards validating and expanding our gene-reaction predictions, glycosyltransferase enzyme activity assays were performed using the

NTP-Glo™ Glycosyltransferase assay (Promega). We used linkage L1:b3GnT and L10:ST6GnT to validate our selections and examined every plausible isoform of the ST3GAL for its ability to perform the linkage L4:ST3GalT reaction. Five acceptors were used: (1) lactose to examine activity on the initial HMO acceptor, (2) LNT and (3) LNnT to establish which enzymes would act on larger type-I and type-II tetrasaccharides, (4) Gal β1,3-GalNAc to determine specificity for non-HMO O-type glycans, and (5) a GlcNAc-β1,3-Gal-β1,4-GlcNAc-β1,3-Gal-β1,4-Glc pentasaccharide structure to test the formation of a non-reducing terminal type-I (Gal-b1,3-) cap on a longer acceptor. We explored the activities of various gene products to perform specific glycosyltransferase reactions crucial to HMO biosynthesis (Fig. 6 and Supplementary Table 3).

In the cross-cohort aggregate analysis (Fig. 5B), B3GNT2 is selected as a reasonable candidate to catalyze flux through the L1:b3GnT reaction. The B3GNT2 support score is nearly 100 times more significant than B3GNT8, the next associated gene. Consistent with the predictions that b3GnT should convert lactose into the precursor to LNT and LNnT, the UDP-Glo™ assay showed B3GNT2 had high activity toward lactose as an acceptor. We further found that B3GNT2 could add a β-1,3-GlcNAc to LNnT as is necessary for poly-lacNAc HMOs. The cross-cohort aggregate analysis (Fig. 5B) selected ST6GALNAC2 to perform L10, the α-2,6 addition of sialic acid to the internal β-1,3-GlcNAc; necessary for the biosynthesis of LSTb from LNT and possibly DSLNT from LSTa. However, the CMP-GLO™ assay highlighted a negligible activity of ST6GALNAC2 toward LNT even at very

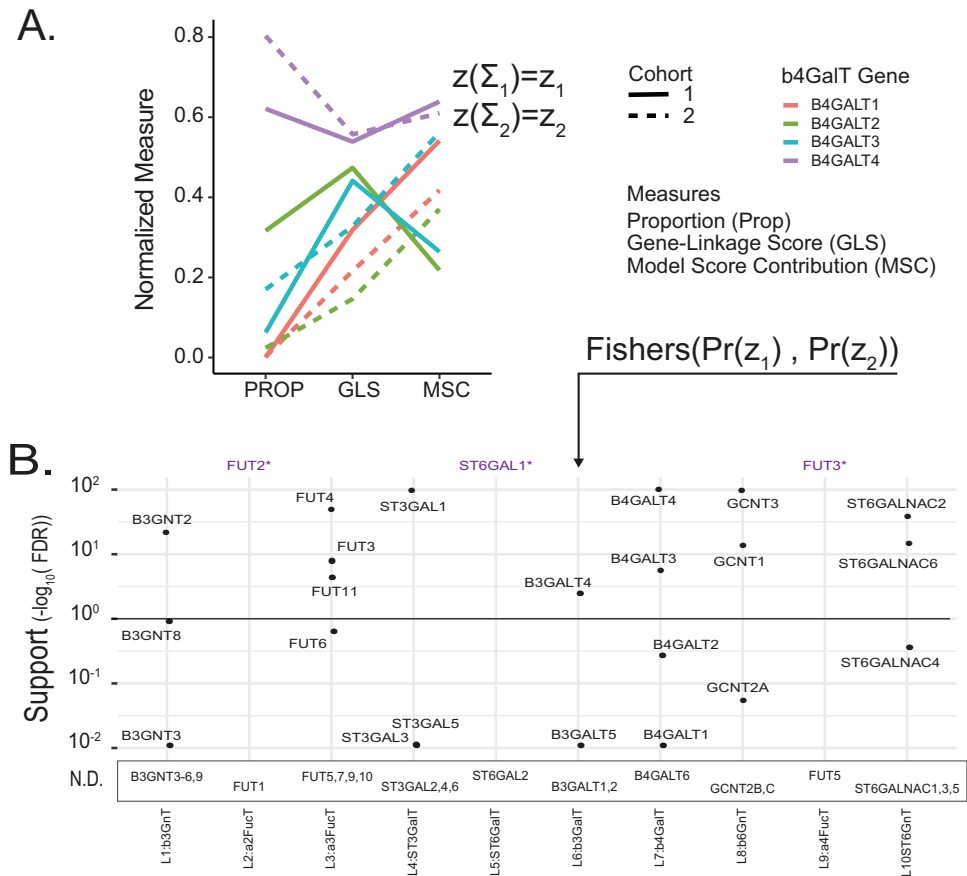

**Fig. 5 Gene expression correlation with model flux predicts enzymes involved in HMO biosynthesis. A** To determine the gene expression that best explains flux through each reaction in each glycomics-transcriptomics matched sample, we examined the proportion of high-performing models were each gene was most flux-correlated (PROP, Supplementary Fig. 11), we also examined the gene-linkage score (GLS) and Model Score Contribution (MSC). Briefly, the GLS measures the correlation between expression of different glycosyltransferase isozymes and normalized flux, whereas the MSC was an enrichment metric assessing the impact of glycosyltransferase isozyme expression on model scores (Supplementary Methods 4.4, 4.5.2, and 5.3). For this visual, each measure was max-min normalized between 0 and 1. Genes were selected based on high performance on all three measures across cohorts (line type). **B** We summarize the three performance scores from panel A across cohorts into a single support score (see Methods section). Briefly, "Support" is derived from the p-value for the sum of PROP, GLS, and MSC z-scores (relative to a permuted background), Fisher-pooled across cohorts then False Discovery Rate (FDR) corrected across genes (see Methods section). Unmeasured genes appear below the plot in the Not Determined (N.D.) box. Genes selected by default (purple, "*") as the only measured gene candidate (Table 1).

high enzyme input indicating that this enzyme does not convert LNT to LSTb. We did not test if it can convert LSTa to DSLNT. In contrast, ST6GALNAC5 was effectively able to use LNT as an acceptor, although we did not confirm the formation of the LSTb structure. ST6GALNAC5 could not be considered in the support score calculation because it was only measured in cohort 2; expression was greater than zero in 1 of 12 samples.

Finally, we tested the affinities of plausible ST3GAL isoforms to sialylate LNT, LNnT or β-1,3-GlcNAc (Supplementary Table 3). The multi-cohort analysis (Fig. 5B) implicates ST3GAL1 as the best candidate for this reaction. The CMP-Glo™ assay indicated that ST3GAL1 has limited activity toward LNT but high activity toward Gal β-1,3-GlcNAc suggesting ST3GAL1, in vitro, is more involved in non-HMO O-type glycan biosynthesis. ST3GAL2 showed a similar but less substantial pattern. ST3GAL3 showed the strongest activity for sialylation both LNT and LNnT suggesting it could synthesize LSTa from LNT. ST3GAL6 shares a similar but lesser activity for LNT and LNnT.

We analyzed the original expression profiles to determine which genes were sufficiently expressed to actuate this activity. STGAL1, 3 and 5 were strongly expressed in nearly 100% of samples across both cohorts; ST3GAL2 and 4 show zero expression in 75% of

samples in at least one cohort (Supplementary Fig. 1). ST3GAL3 was highly expressed and effective at catalyzing the L4 reaction for LNT and LNnT while ST3GAL1 was highly expressed and weakly catalyzed sialylation of LNT making ST3GAL3 the most likely candidate for L4 reaction on LNT and LNnT.

**Selected glycosyltransferases share transcriptional regulators across independent predictions.** To explore the transcriptional regulation during lactation, we used two orthogonal approaches for transcription factor (TF) discovery. We used Ingenuity Pathway Analysis (IPA) to predict upstream regulatory factors based on differential expression (DE) associated with each HMO. IPA analyzed all genes differentially expressed with HMO abundance, not only HMO glycogenes; these DE patterns formed HMO-specific gene expression signatures. Additionally, we used MEME for de novo motif discovery in the promoter regions of HMO glycogenes and TOMTOM to map those discovered motifs to known TFs. We validated these predictions by examining transcriptional regulators selected by both MEME and IPA (Supplementary Figs. 16–S22, see Methods section).

IPA discovered 57 TFs significantly ($|z| \geq 3$; $p < 0.001$) associated with the 16 HMO-specific gene expression signatures. We

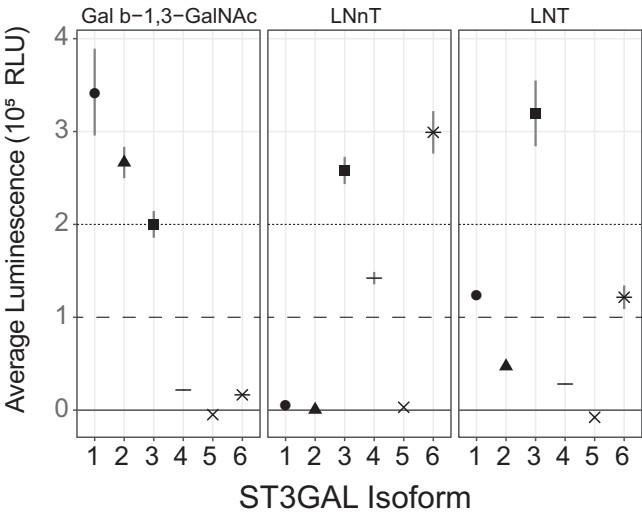

**Fig. 6 Results of the CMP-Glo™ Glycosyltransferase Assay to test GT candidates on relevant HMO acceptors.** Average luminescence below 10,000 is considered weak activity, and activity above 200,000 is considered very high activity. Reported luminescence values were background corrected and 95% confidence intervals are shown. For complete details see Supplementary Table 4. Shapes correspond to ST3GALT isoforms.

performed DE on HMO substructure abundances and substructure abundance ratios[17]; IPA found 66 and 49 TFs significantly ($|z| \geq 3$; $p < 0.001$)) associated with HMO substructure and substructure ratio specific gene expression signatures. Using MEME, we identified three putative TF regulatory sites (TF motifs I, II, and III) for 6 selected glycosyltransferases responsible for the HMO biosynthesis (Table 2 and Supplementary Fig. 18). TOMTOM calculated that these putative binding sites were significantly associated with six known TFs (IKZF1, SP1, EGR1, ETS1, ETV4, and ERG) that were also predicted by IPA as regulators of gene signatures associated with HMO concentration (Fig. 7 and Supplementary Fig. 19) or HMO glycan substructures abundance (Supplementary Fig. 20). SP1, EGR1, ETS1, ETV4, and ERG are all predicted to positively influence expression associated with the biosynthetically related HMOs: 3′SL, 3FL, LSTb, and DSLNT; 3′SL and 3FL share a common substrate (lactose) while LSTb is a likely precursor to DSLNT. The motif-level analysis showed opposing regulation between IKZF1: upregulating gene expression signatures associated with the 3′SL and LSTb substructure abundance[17] (X34 and X62 respectively, see Supplementary Figs. 20, 21) and down-regulating gene expression associated with GlcNAC-lactose, LNT and LNFPI substructure abundance (X18, X40, and X65 respectively, see Supplementary Figs. 20, 21), while EGR1, ERG, and ETS1 have the opposite predicted impact (Supplementary Fig. 20). The motif-level predictions are consistent with the HMO-level predictions of upregulation on 3′SL and LSTb while adding an additional point of contrast. While EGR1, ERG, and ETS1 are predicted to increase production of sialylated HMOs, they may have the opposite impact on LNFPI. Thus, we detect signatures of multiple transcription factors that could coordinate the regulation of the genes we identified to contribute to HMO biosynthesis (see Supplementary Discussion).

## Discussion
By integrating sample-matched quantitative oligosaccharide measurements and gene expression data using computational models of HMO biosynthesis, we resolved genes responsible for 10 elementary reactions in human mammary gland epithelial cells. The

biosynthetic model is essentially a probabilistic model where each node represents a specific glycan structure, each edge a known possible enzymatic reaction converting one glycan to another, and edge weight the possibility of such a conversion. This method is highly efficient and intuitive for the iterative HMO biosynthetic network construction due to the modular nature of monosaccharide addition to existing glycan structures during HMO molecular extensions[13,36]. Compared to well-known (i.e., FUT2 & ST6GAL1) events and empirically validated (confirmed by kinetic assays and expressed in milk), we observed 4 true-positive, 7 true-negative, 1 false-positive and 1 false-negative gene-enzyme prediction using our approach (sensitivity = 0.875, specificity = 0.8, positive-predictive value = 0.875, precision = 0.875, recall = 0.875). Our approach correctly resolved both well-known reactions. Kinetic assays showed our approach selected milk-expressed substrate accepting gene-enzyme pairs for reactions L1 and L4 but not L10. Kinetic assays also found a false-negative prediction for reaction L4 and a false-positive prediction for reaction L10.

In comparison with traditional kinetic models of glycan synthesis, the low-parameter framework can utilize either LC or MS data and also allowed inference of enzymatic activities[36] using model parameters, which could be readily validated with the transcriptomics data of involved glycosyltransferases. For mapping out the pathways, we also took a different approach compared to existing retrosynthesis approaches[37–44]. While there are strengths with the retrosynthesis approaches, our model allowed us to account for promiscuity of glycosyltranferases in the synthesis of the oligosaccharides. It also allowed us to readily analyze transcriptomics data in the pathways and further quantitatively differentiate between candidate isozyme enzymatic activities. The modeling-based strategy was essential since simple correlations failed to capture the simplest HMO-gene associations, given the complex interactions of glycosyltransferases in the HMO biosynthetic pathway. Because the pathway characterization is still incomplete, we built >44 million candidate models that uniquely recapitulate glycoprofiling data in two independent cohorts. Candidate model flux, i.e., activity of each reaction, was predicted for each model and compared to sample-matched gene expression data. We used the consistency between gene expression and predicted flux across cohorts in high-performing models to select genes for each fundamental reaction. Analysis of these models suggested glycosyltransferase genes, thus providing a clearer picture of the enzymes and regulators of HMO biosynthesis in mammary epithelial cells. The clarification of the pathways and enzymes involved in HMO biosynthesis will be an invaluable resource to help (1) discover the maternal genetic basis of health-impacting[1,2,5,6,45–54] HMO composition heterogeneity[7,12,25,55] and (2) drive chemoenzymatic synthesis[56–60] and metabolic engineering for manufacturing HMOs for food ingredients, supplements and potential therapeutics[61–66] (see Supplementary Discussion).

Of the three fucosylation reactions, two were effectively determined using expression data alone while the third required additional insight from the flux-expression comparison or support score. Consistent with studies in blood[22–24] and milk[25,55,67] types, we selected FUT2 as the gene supporting the α-1,2-fucosylation (L2:a2FucT) linkage reaction. FUT1 was ruled out due to non-expression (Supplementary Results, Supplementary Dataset 2). In the second fucosylation reaction, FUT3, FUT4 and FUT11 all show significant support for α-1,3-fucosylation (L3:a3FucT) linkage formation. FUT11 is more commonly considered an *N*-glycan-specific transferase[68] and therefore a less likely candidate. Both FUT3 and FUT4 prefer to fucosylate the inner GlcNAc of a type-I polylactosamine[69]. FUT3 prefers neutral type-I polylactosamine while FUT4 also fucosylates the sialylated form[70,71]; the charge preferences are inverted for type-II polylactosamine acceptors[72]. Prudden et al.[59] used FUT9 to perform this reaction, consistent with its ability to transfer α-1,3-fucose to the distal GlcNAc of a neutral

**Table 2 TF motif (MEME) and IPA upstream regulator integrated results.**

| Linkage | Reaction support score selected candidate | MEME TF Motif (Mixture model likelihood[124] *p*-value)[a] | JASPAR TF (Motif enrichment[126] *p*-value)[b] | IPA predicted TF[c] |
|---|---|---|---|---|
| L1:b3GnT | B3GNT2 | TF Motif–II (1.39e-12) | SP1 (4.96e-05) | Y |
| | | | EGR1 (2.17e-05) | Y |
| L2:a2FucT | FUT2 | TF Motif–III (3.63e-16) | IKZF1 (7.62e-04) | Y |
| L3:a3FucT | FUT11 | TF Motif–II (1.00e-7) | SP1 (4.96e-05) | Y |
| | | | EGR1 (2.17e-05) | Y |
| L4:ST3GalT | ST3GAL1 | TF Motif–I (2.76e-11) | ETV4 (1.24e-03) | Y |
| | | | ETS1 (3.01e-04) | Y |
| | | | ERG (3.50e-04) | Y |
| L7:b4GalT | B4GALT4 | TF Motif–II (7.67e-11) | SP1 (4.96e-05) | Y |
| | | | EGR1 (2.17e-05) | Y |
| L10: ST6GnT | ST6GALNAC2 | TF Motif–II (1.08e-7) | SP1 (4.96e-05) | Y |
| | | | EGR1 (2.17e-05) | Y |

[a]The *p*-value (see Supplementary Fig. 18) is the significance of the selected GT to the MEME identified TF motif.
[b]The *p*-value (see Supplementary Table 5) is the significance of known TF associated with the MEME identified TF motif.
[c]The IPA upstream regulator analyses were conducted on the three different sets of DEGs: 16 HMOs, 19 glycan motifs, and 4 differential motifs (see Methods section). Based on the *Z*-score predicted by IPA using the gene expression data, we selected the significant TFs with IPA predicted activation score |*Z* value|> = 3 in this study. Note, 'Y' denotes the known TF is presented in the indicated dataset (HMO (Fig. 7 and Supplementary Fig. 19), Motif (Supplementary Fig. 20), or differential motif (Supplementary Fig. 22)) of the IPA predicted TF and 'N' means the TF does not present in the dataset of IPA predicted TF.

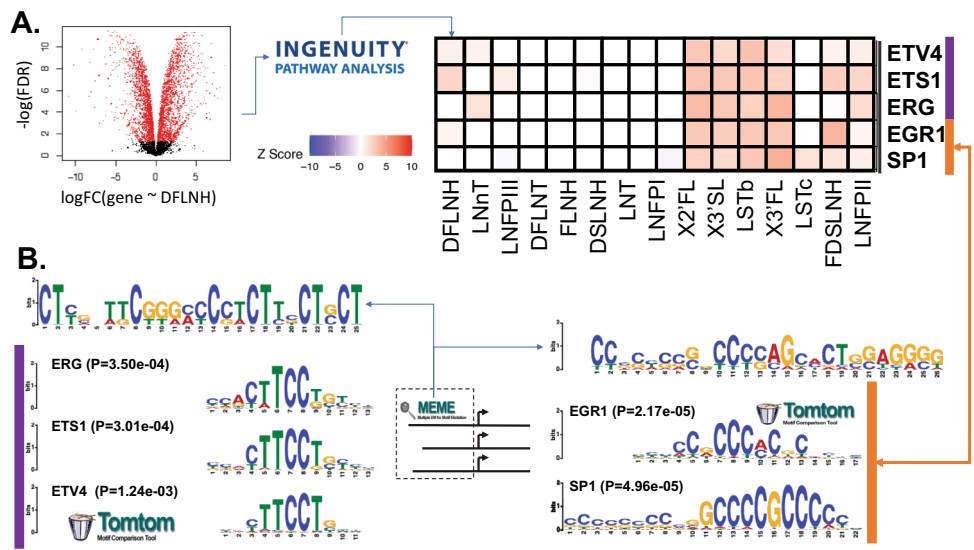

**Fig. 7 de novo promoter-enriched TF motifs and IPA predicted TFs using differential expression analyses with respect to 16 HMOs. A** MEME identified TF motifs and 5 known TFs (ETV4, ETS1, EGR1, SP1, and ERG) associated with them (see Supplementary Table 4). MEME-discovered TFs were cross-referenced with known TF binding sites using TOMTOM. Logos for the matched known and discovered motifs are shown in the top and bottom of each subpanel; the *p*-value is a logo matching significance calculated by TOMTOM. **B** Subset of a biclustering of activation z-score computed by IPA indicating the likelihood that a TF activates (z > 0) or inhibits (z < 0) an HMO concentration signature (gene expression associated with changes in HMO concentration). The full biclustering can be found in the supplement (Supplementary Fig. 19).

polylactosamine[68–70]. The four HMO structures with α-1,3-fucose in the summary network (Fig. 4) include 3FL (neutral inner fucosylation), LNFPIII (neutral distal fucosylation), DFLNT2 (neutral inner fucosylation), and FDSLNH2 (sialylated and neutral distal fucosylation). FUT9 showed negligible expression in RNA-Seq (3rd Quartile TPM = 0.37, Supplementary Dataset 2), yet it is highly expressed (TPM > 10) brain and stomach[31]. Therefore, it is more likely that the distal fucosylation is conducted by another enzyme in vivo while the inner fucosylation is likely performed by either FUT3 or FUT4. FUT3 was also implicated for the α-1,4-fucosylation (L9:a4FucT) by default due to the FUT5 non-expression; FUT5,

selected by Pruden et al. for L9:a4FucT, was expressed in milk sample RNA-Seq (Supplementary Results, Supplementary Dataset 2). Therefore, FUT5 can neither be evaluated nor dismissed as a candidate gene. FUT3 adds an α-1,4-fucose to the GlcNAc of a neutral type-I chain to form the Lewis-A or Lewis-B group and adds an α-1,3-fucose to the GlcNAc of a type-II chain[70,71]. Usage of FUT3 would provide a parsimonious explanation for the fucosylation of both type-I and type-II HMOs like LNFPII (Fuc-α-1,4-LNT (type-I)) and LNFPIII (Fuc-α-1,3-LNnT (type-II)).

One of two sialyltrasferases was clearly resolved with expression data alone, the other required additional examination.

ST6GAL1 was chosen by default to support the α-2,6-sialylation (L5:ST6GalT) reaction due to the non-expression of ST6GAL2 (Supplementary Dataset 2). ST6GAL1 sialylates galactose in HMOs[59]. For the second sialylation reaction, our flux-expression comparison selected ST6GALNAC2 and ST6GALNAC6 as the significant supporters of α-2,6-sialylation (L10:ST6GnT). Through a kinetic assay, we found that ST6GALNAC2 (previously shown to accept core-1 O-glycans[73,74]) fails to sialylate LNT. Though our kinetic assay shows that ST6GALNAC5 (known to sialylate GM1b[75]) can sialylate LNT, it was not expressed in this context (Supplementary Results, Supplementary Dataset 2). ST6GALNAC3 expression was not observed in microarrays but could not be ruled out due to RNA-Seq expression (Supplementary Dataset 2, Supplementary Results); it sialylates the GalNAc of NeuAc-α-2,3-Gal-β-1,3-GalNAc-α-1-O-Ser/Thr and NeuAc-α-2,3-Gal-β-1,3-GalNAc-β-1,4-Gal-β-1,4-Glc-β-1-Cer when the inner galactose is not sialylated (e.g. GD1a or GT1b)[76–79] but has not been shown to transfer to a GlcNAc. The last ganglioside-accepting family gene, ST6GALNAC6, has broader activity accepting several gangliosides (GM1b, GD1a, and GT1b)[76] and sialylating the GlcNAc of LNT-ceramide[80]. Considering the broader activity, clear expression, and computational selection, ST6GALNAC6 is the most likely candidate, though ST6GALNAC3 should not be ruled out. In the third reaction, ST3GAL1 shows significant support for α-2,3-sialylation (L4:ST3GalT) reactions while ST3GAL3 shows negligible consistency in the flux-expression comparison. Yet, in vitro, ST3GAL3 was most effective at sialylating both LNT and LNnT in kinetic assays while ST3GAL1 weakly sialylated LNT. ST3GAL4, which prefers type-II acceptors[81–83], was used previously to perform this reaction in vitro[59], but it was not expressed on the microarrays nor RNA-Seq. ST3GAL3 can accept type-I, type-II, and type-III acceptors including LNT and prefer type-I acceptors[81,82,84] while ST3GAL1 accepts type-I, type-III, and core-1 acceptors but not type-II[81,82,85]. The kinetic assays and previous literature show ST3GAL3 is more capable than ST3GAL1 at catalyzing this reaction, while ST3GAL1 expression was found to be the only plausible candidate based on estimated flux through this reaction. If ST3GAL1 were responsible for this reaction, its inability to sialylate type-II HMO could partially explain the lack of sialylation and larger structures in the type-II HMO branch. Both ST3GAL1 and ST3GAL3 remain plausible candidate genes, and further in vivo studies are needed.

Both galactosylation reactions required further examination of flux-expression relationships. We found B3GALT4 to significantly support the type-I β-1,3-galactose addition (L6:b3GalT). B3GALT4 can transfer a galactose to GalNAc in the synthesis of GM1 from GM2[86]. Unlike B3GALT5, there is no evidence that B3GALT4 can transfer galactose to a GlcNAc[87]. B3GALT5, has been shown to transfer a β-1,3-galactose to GlcNAc to form LNT in vitro[88]. B3GALT5 expression measured for cohort 1 microarray was much lower than expression in cohort 2 and the independent RNA-Seq[30] suggesting that the probes in the first microarray may have failed (Supplementary Dataset 2 and Supplementary Results). While both B3GALT4 and B3GALT5 seem plausible, given the historical failures of B3GALT4 to perform this reaction and our likely failure to measure and evaluate B3GALT5, B3GALT5 may be the stronger candidate for this reaction. In the second galactosylation reaction, the flux-expression comparison found B4GAL4 and B3GALT3 most significantly supports the type-II definitive β-1,4-galactose addition (L7:b4GalT). These gene-products can synthesize LNnT-ceramide[89]. Additionally, in the presence of α-lactalbumin (highly expressed during lactation), B4GALT4 shows an increased affinity for GlcNAc acceptors suggesting during lactation it is more likely to perform the L7 reaction[89,90]. B4GALT1 and B4GALT2 synthesize lactose in the presence of α-lactalbumin during lactation[34,35], but

B4GALT1 expression was not correlated with L7 flux and B4GALT2 was not expressed (Supplementary Dataset 2). Associations between B4GALT1 expression L7 flux may be masked due to its consistent high. Therefore, flux-expression correlation should not be used to exclude B4GALT1 as a candidate for the L7 reaction. Doing so, B4GALT4, B4GALT3 and possibly B3GALT1 remain the most plausible candidates.

Finally, both GlcNAc additions required flux-expression examinations. B3GNT2 showed significant support in the flux-expression comparison. In our kinetic assays, B3GNT2 demonstrated high activity towards lactose as an acceptor. Previously, B3GNT2 has performed the β-1,3-GlcNAc addition (L1:b3GnT) on multiple glycan types including several HMOs: lactose, LNnT, polylactosamine-LNnT[91]. The agreement of literature, kinetic assays, and flux-expression analysis indicate B3GNT2 is an appropriate choice for this reaction. In the second GlcNAc reaction, GCNT3 and GCNT1 most significantly support the branching β-1,6-GlcNAc addition (L8:b6GnT). While GCNT2B can effectively transfer the branching GlcNAc to the inner galactose of LNnT[59,92], it was not expressed in the cohort microarrays nor independent RNA-Seq. GCNT1 transfers a branching GlcNAc to the GalNAc of a core-1 O-glycan[93,94] while GCNT3 acts on core-1 and the galactose of the LNT-like core-3 structure[94,95]. GCNT3 is also specifically expressed in mucus-producing tissues[94,95] like lactating mammary gland epithelium. Interestingly, GCNT3 acts on galactose of the GlcNAc-β-1,3-Gal-β-1,4-Glc trisaccharide (predistally) while GCNT2 acts on the central galactose of the LNnT or LNT tetrasaccharide (centrally)[92]. Therefore, reliance on GCNT3 for the branching reaction would explain the noncanonical branched tetrasaccharide (HMO8, Fig. 4) suggesting a third major branch from GlcNAc- β-1,6-lactose, distinct from LNT and LNnT. Predistal addition of the branched GlcNAc may also explain the lack of branched type-II structures since B4GALT4 cannot act on branched core-4 structures[96]. HMO biosynthesis with GCNT3 and B4GALT4 could explain the type-I bias seen in the summary network (Fig. 4).

We note that our approach relies on several simplifying assumptions. Well-chosen assumptions can increase generalizability of a model while mitigating overspecification[97]. While previous models have demonstrated the importance of defining subcellular compartmentalization of individual steps of glycosylation[13] and sugar nucleotide availability[98], we found our models could recapitulate the HPLC data without such specifications. However, we anticipate that further study of sugar nucleotide concentrations and glycosyltransferase expression at the single-cell level[99] will result in additional insights into the regulation of HMO biosynthesis. While for such studies, further data will be needed, this study here lays the groundwork by resolving gene-protein-reaction relations underlying HMO biosynthesis.

Our results show consistency with experimental validation here and the published literature. Further direct empirical studies will be invaluable to confirm each gene-reaction association and the complete biosynthesis network. Such studies would include further clinical cohort studies and the development of mammary organoid models capable of producing HMOs. Such experimental systems can clarify the impact of mammary-tissue specific genes, cofactors, and HMO chaperones like α-lactalbumin[89,90] on glycosyltransferase activity. Therefore, further development of authentic in vitro cell and organoid models will be invaluable to finalizing our model of HMO biosynthesis.

By using systems biology approaches, different omics data can be integrated, as shown here to predict gene-reaction relations even in highly uncertain and underdetermined networks. Of the ten fundamental reactions we aimed to resolve and reduce (Table 1), we succeeded in narrowing the candidate substantially

for each one. The newly reduced space of HMO biosynthetic pathways and knowledge of the enzymes and their regulation will enable mechanistic insights into the relationship of maternal genotype and infant development. Finally, once essential HMOs are identified, the knowledge presented here on the HMO biosynthetic network can provide insights for large-scale synthesis of HMOs as ingredients, supplements, or potential therapeutics to further help improve the health of infants, mothers, and people of all ages.

## Methods

**Milk sample collection**. Samples were collected following Institutional Review Board approval (Baylor College of Medicine, Houston, TX). Lactating women 18–35 years of age with uncomplicated singleton pregnancy, vaginal delivery at term (>37 weeks), Body Mass Index <26 kg/m² without diabetes, impaired glucose tolerance, anemia, or renal or hepatic dysfunction were given informed consent before sample collection. Description of the protocols used to collect milk samples and the diversity of subjects present in both datasets. Cohort 1 consists of 8 samples for each of the 6 subjects (48 samples total) including milk from 4 secretor mothers and 2 non-secretor mothers spanning from 6 h to 42 days postpartum. Cohort 2 consists of 2 samples over each of the 5 (10 samples total) including samples from 4 secretor mothers and 1 non-secretor mother spanning 1 to 2 days postpartum. Data from cohort 1[27,28] and cohort 2[29] have been previously published and comprehensively described.

**Illumina mRNA microarrays and glycoprofiling**. All expression and glycoprofiling measurements were sample-matched. Therefore, comparisons across datatypes occurred within each individual sample described in the previous section. Not all samples in these studies have both microarray and glycoprofile measurements, only the samples described in the previous section have matched glycomics and transcriptomics data.

mRNA was isolated from TRIzol-treated milk fat in each sample. Expression in cohort 1 was measured using HumanHT-12 v4 Expression Beadchip microarrays (Illumina, Inc.) with ~44k probes. Gene expression data for cohort 1 were retrieved from the Gene Expression Omnibus at accession: GSE36936 [ncbi.nlm.nih.gov/geo/query/acc.cgi?acc=GSE36936]. Cohort 2 gene expression data were measured using a Human Ref-8 BeadChip array (Illumina, Inc) with ~22k probes. Expression data for cohort 1 can be accessed at accession: GSE12669 [ncbi.nlm.nih.gov/geo/query/acc.cgi?acc=GSE12669]. Both microarrays were background corrected. The cohort 1 microarray was normalized using cubic spline normalization and the cohort 2 microarray was normalized using the robust spline normalization.

As previously described[100], HMO absolute concentrations, nmol/mL, based on HMO standard response curves and internal standard concentrations. Specifically, HMO concentration data were collected using high-performance liquid chromatography (HPLC) with 2-aminobenzamide (CID: 6942) derivatization. To measure absolute concentration, samples were spiked with a non-HMO carbohydrate, a raffinose (CID:439242) as an internal standard. 16 HMOs were measured using retention time and commercial standards including 2-fucosyllactose (2′FL), 3-fucosyllactose (3FL), 3-sialyllactose (3′SL), lacto-N-tetraose (LNT), lacto-N-neotetraose (LNnT), lacto-N-fucopentaose (LNFP1, LNFP2 and LNFP3), sialyl-LNT (LSTb and LSTc), difucosyl-LNT (DFLNT), disialyllacto-N-tetraose (DSLNT), fucosyl-lacto-N-hexaose (FLNH), difucosyl-lacto-N-hexaose (DFLNH), fucosyl-disialyl-lacto-N-hexaose (FDSLNH) and disialyl-lacto-N-hexaose (DSLNH). Technicians were blinded to sample metadata. HMO compositions and the absolute abundance measurements for cohort 1 were fully described[100]. HPLC quantification of HMO data used Chromeleon v7.2. The relative abundance of each glycan in each milk sample is normalized by the total absolute abundance of the 16 most abundant (typically >95% of HMO mass per sample[100]) HMO signals (chromatogram signals) for a given sample when used for model construction/fitting, as described by Bao et al.[17]. Measurements for cohort 2 were previously unpublished but used the same methodology as cohort 1. Relative HMO abundance was calculated as the ratio of each HMO weight to the total HMO weight for a sample. Raw HMO concentrations and normalized glycogene expression is provided in Supplementary Dataset 1.

**Software**. Modeling of HMO biosynthesis was performed in Matlab 2016b using the CobraToolbox v3[101]. All analysis of biosynthetic models, interpretation and statistics were performed in R v3.6. In R, we used *bigmemory v4.5.36, bigalgebra v1.0.1 and biganalytics v1.1.21* to handle the millions of models and associated statistics[102]. We used *metap* for pooling p-values[103].

**Curation of undetermined HMO structures from literature**. Of the many possible HMOs, more than 150 have been identified (Ninonuevo 2006; Wu, 2010; Wu, 2011) and several of the most abundant observed HMOs remain to have ambiguous structures. The natural heterogeneity (branching, isomerization and polarization) of HMO mixture present in milk makes their structural identification and quantitative detection a prohibitive challenge to many current studies[6,8,104]. This is

due in part to the lack of standard for performing a comprehensive study and spectral characterization of each HMO. In this study, we analyzed 16 of the most abundant HMOs, 11 of which have fully determined molecular structure, while the remaining five have multiple alternate candidate structures[8]. We were very careful throughout the paper to distinguish evidence-supported isomeric HMO structures and to present the possible structures based on known reaction rules (Supplementary Figs. 5, 14, and 21).

**Generation and scoring of glycosylation networks models**. Here we attempt to determine the genes responsible for making HMOs through the construction and interrogation of models of their biosynthesis. Similar to the other biosynthetically constrained glycomic models like the milk metaglycome[21], Cartoonist[94], and several N-glycome simulations[13,105–107], we began with a set of elementary reactions. Enumerating all feasible permutations of the elementary reaction (Fig. 3A and Supplementary Methods 4.1), we delineated every possible reaction series from lactose to each of the 16 measured HMOs. Of the measured HMOs, 11 have fully determined molecular structures, while the remaining five have multiple candidate structures (Fig. 1C and Supplementary Fig. 5)[6,8,33,104,108,109]. The set of all possible reactions leading to characterized and ambiguous structures formed the Complete Network (Fig. 3B and Supplementary Methods 4.1). Though non-lysosomal glycosidase[110–112] reactions are not explicitly specified, they are implicitly encoded in the flux. To reduce the Complete Network to a more manageable size, we identified and removed all reactions that do not lead to observed oligosaccharides using Flux Variability Analysis (FVA; Supplementary Methods 5.4;[113–115]). This trimming (Fig. 3C and Supplementary Methods 4.2) defines the Reduced Network (Fig. 3D and Supplementary Methods 4.2). The Reduced Network describes many candidate models that can uniquely simulate the HMO abundance collected through High-Performance Liquid Chromatography (HPLC). A Mixed Integer Linear Programing (MILP, Supplementary Methods 5.5; refs. [116,117]) approach was employed to extract candidate models from the Reduced Network capable of uniquely recapitulating the HPLC data with a minimal reaction set (Fig. 3E; Supplementary Methods 4.3; Supplementary Dataset 1 - Raw HPLC & Microarray). The reactions of each candidate model were parameterized to determine the necessary flow of material (flux) through each reaction to reproduce the measured oligosaccharide profiles (Fig. 3F and Supplementary Methods 4.3, 5.5). The models were ranked by the consistency between the predicted flux and the expression of genes believed to be associated with each reaction (Fig. 3G and Supplementary Methods 4.4). This consistency is evaluated by the Spearman correlation of changes in flux and gene expression across subjects (Fig. 3H and Supplementary Methods 4.4.1).

**Candidate model ranking, model selection, and selection validation**. Model scores, indicating the consistency between flux and gene expression (Supplementary Methods 4.4.1), were used to rank candidate models (Supplementary Methods 4.4.2). The distribution of model scores computed from each dataset were approximately normal, as evidenced by their linear Q-Q plots. This permitted the construction of a background normal distribution of model scores (Supplementary Fig. 6). We then selected high-performing models, those with z-score normalized model scores greater than 1.646 (i.e., greater than the top 5% of scores from a normal distribution) for further study. The model score threshold was varied from 4–8% to establish robustness in the results; subsequent analyses were negligibly sensitive to this threshold. Model selection was performed on scores computed independently for cohort 1 and cohort 2. Commonly high-performing models were those that performed well in both cohort 1 and cohort 2. Hypergeometric enrichment was used to confirm that the top cohort 1 and cohort 2 models significantly overlapped (see Supplementary Methods 4.4.2).

**Summary network extraction from the Reduced Network**. The summary network relates a heuristic selection of the most important reactions in the HMO biosynthesis network as measured by proportion of inclusion in the commonly high-performing models and enrichment in the commonly high-performing models relative to the background. Paths drawn from observed HMOs to the root lactose were scored for their aggregate importance. The top 5% of paths leading to each observed HMO were retained to form the summary network (see Supplementary Methods 4.4.3).

**Ambiguous gene selection**. We aimed to match 10 elementary glycosyltransferase reactions to the supporting genes (Table 1). Candidate genes were filtered from the relevant gene families to exclude gene products well known to perform unrelated reactions (Table 1). Candidate genes were first evaluated for expression in breast epithelium samples including microarrays in this study, independent RNA-Seq (GSE45669)[30] and comparison to global expression distributions in GTEx[31]; genes unmeasured in at least 75% of microarray samples (3rd Quartile, Q3) within each cohort were excluded unless they were non-negligibly expressed in the independent RNA-Seq ($TPM_{Lemay} > 2$ or $TPM_{Lemay} > Median(TPM_{GTEx})$ (see Supplementary Results, Supplementary Dataset 2, Supplementary Fig. 4).

We used the model score definition, which quantifies how well the genes explain a model - i.e., if the expressions of the genes are best correlated to the normalized flux of the reaction (Supplementary Fig. 7 and Supplementary

Methods 4.4) they are proposed to support. We examined each gene contribution to the overall model score in three ways to determine a consensus support score for each gene-reaction association (see Supplementary Methods 4.5.2).

The first metric we examined was the proportion (PROP) of commonly high-performing models best explained by an isoform relative to the proportion of background models that select that same isoform. The second metric was the average gene-linkage score (GLS) in high-performing models - i.e., the Spearman correlation between the normalized flux (Supplementary Fig. 7 and Supplementary Methods 4.4) and gene expression of corresponding candidate genes. The gene-linkage score is a continuous measure of the consistency between each gene with the flux it was proposed to support. Because it considers every gene, not just the most flux-consistent gene, it is helpful for judging performance when the most flux-consistent gene is more ambiguous. The third metric was the model-score contribution (MSC). MSC quantifies the Pearson correlation between the gene-linkage score, the gene expression consistency with the normalized flux, and the overall model score (i.e., the average correlation of all most-flux-consistent genes). The model score indicates the frequency with which a gene is the most flux-consistent gene normalized by its contribution relative to the other most flux-consistent genes in that model.

An aggregate reaction support score was constructed to describe performance within each individual score (PROP, GLS, and MSC) and consistency across cohorts. To measure significance, the gene-linkage score matrix (i.e., Spearman correlation between each candidate gene and the corresponding normalized flux for each model) was shuffled (n = 27) and all analyses rerun on each shuffle to generate a permuted background distribution for PROP, GLS, and MSC; shuffling of the GLS matrix was done using a perfect minimal hash to remap all entries back to the GLS matrix in a random order[118]. Performance within each independent cohort was described as the sum of z-scores for each of three measures; z-score was calculated relative to the mean and standard deviations of these scores in the permutation results. Consistency across cohorts was determined by pooling p-values using the Fisher's log-sum method[103,119]. The score presented in Fig. 5B is the -log$_{10}$(FDR(cohort-pooled-p).

**In vitro glycosyltransferase activity assays.** Recombinant forms of the respective glycosyltransferases were expressed and purified as previously described[120]. Enzyme activity was determined using the UDP-Glo$^{TM}$ or UMP/CMP-Glo$^{TM}$ Glycosyltransferase Assay (Promega) that determined UDP/CMP concentration formed as a by-product of the glycosyltransferase reaction. Assays were performed according to the manufacturer's instructions using reactions (10 μL) that consisted of a universal buffer containing 100 mM each of MES, MOPS, and TRIS, pH 7.0, donor (1 mM UDP-GlcNAc (Promega) for B3GNT2; 1 mM UDP-Gal (Promega) for B3GALT2; 0.2 mM CMP-SA (Nacalai USA Inc.) for ST3GAL1-6, ST6GAL-NAC2, and ST6GALNAC5), 1 mM acceptor (lactose (Sigma) and lacto-*N*-neotetraose (LNnT) (Carbosynth) for B3GNT2; lacto-*N*-tetraose (LNT, Bode lab) and pentasaccharide (GlcNAc-b1,3-Gal-b1,4-GlcNAc-b1,3-Gal- b1,4-Glc, Boons lab, University of Georgia) for B3GALT2; LNnT, LNT, and Gal-β1,3-GalNAc (Carbosynth) for ST3GAL1-6; LNT for ST6GALNAC2 and ST6GALNAC5. The B3GNT2 and B3GALT2 assays also contained 1 mg/ml BSA and 5 mM MnCl$_2$. Assays were carried out for 1 h (B3GNT2, B3GALT2, ST6GALNAC2, and ST6GALNAC5) or 30 min (ST3GAL1-6) at 37 °C. Reactions (5 μL) were stopped by mixing with an equal volume of Detection Reagent (5 μL) in white polystyrene, low-volume, 384-well assay plates (Corning) and incubated for 60 min at room temperature. After incubation, luminescence measurements were performed using a GloMax Multi Detection System plate reader (Promega). The average luminescence was subtracted from the average luminescence of the respective blank to correct for background. Background and reaction measurements were performed in triplicate.

**Differential expression analysis.** The DE analysis was conducted on three different datasets: (1) 16 different HMOs (2′FL, 3′SL, 3FL, FLNH, LNT, LNnT, LSTb, LNFP-III, LNFP-II, LNFP-I, DFLNT, LSTc, DSLNT, FDSLNH, DSLNH, DFLNH), (2) 19 glycan motifs (X18, X32, X34, X35, X37, X40, X62, X63, X64, X65, X66, X94, X106, X113, X120, X127, X141, X142, X143, see Supplementary Figs. 21, 22), and (3) 4 differential motifs for the difference ("conversion rate") between related motifs (X65-X40, X106-X62, X63-X37, X62-X40, see Supplementary Fig. 21, 22). Substructure abundance for glycan motifs and conversion ratios were computed using Gly-Compare v1[17]. The gene expression data were downloaded from the Gene Expression Omnibus[121] (GSE36936). Specifically, for each HMO, motif or differential motif, we used concentration (e.g., HMO–*3FL*) as the predictor for gene expression in the DE analysis (e.g., "gene expression ~ [3FL]"). The DE analysis was performed by fitting linear models using the empirical Bayes method as implemented in the *limma* v3.40.6 in R v3.6.1 package[122] and p-values were adjusted for multiple testing using Benjamini-Hochberg (BH) method[123]. In this way, we determined gene-expression signatures indicative of each HMO and motif abundance.

**Ingenuity pathway analysis upstream regulator analysis.** DE signatures indicative of differential abundance in 16 HMOs, 19 motifs, and 4 differential motifs were analyzed to predict upstream regulators using Ingenuity Pathway Analysis (IPA, QIAGEN Inc.). Gene expression signatures indicative of HMO and motif

abundance were defined as genes differentially expressed with abundance in the previous *limma* analysis (FDR q < 0.05 and |Fold Change|>1.5).

**de novo TF binding site motifs discovery and known TF binding site identification.** We downloaded promoter sequences (file: upstream1000.fa.gz; version: GRCH38) from UCSC Genome Browser public database (https://genome.ucsc.edu/) for the O-glycosyltransferase genes used in this study (Supplementary Dataset 2). These promoter sequences included 1000 bases upstream of annotated transcription starts of RefSeq genes with annotated 5′ UTRs. To conduct de novo TF binding site motifs discovery, we first applied the motif discovery program MEME[124] to identify candidate TF binding site motifs on the downloaded promoter sequences with default parameters. The 10 TF binding site motifs found by MEME were analyzed further for matches to known TF binding sites for mammalian transcription factors in the motif databases, JASPAR Vertebrates[125], via motif comparison tool, TOMTOM[126]. The resulting discovered TF binding site motifs and their significantly associated known TF binding sites (Supplementary Tables 4, 5) for mammalian transcription factors were used further to compare with the IPA-predicted upstream regulators.

**Reporting summary.** Further information on research design is available in the Nature Research Reporting Summary linked to this article.

## Data availability
All raw data (glycan abundance and processed expression data) have been deposited in GitHub [https://github.com/bkellman/HMO_GeneReaction_pred]. All intermediary data (including models, flux, correlation, gene linkage scores, gene/structure proportions, and model scores) have been deposited in Zenodo (https://doi.org/10.5281/zenodo.4060217). Data generated for this study and processed data are provided in the Supplementary Information/Source Data file. All data are available under a CC-BY-4.0 license. Source data are provided with this paper.

## Code availability
All code used to conduct this work is available: github.com/LewisLabUCSD/HMO_GeneReaction_pred Reference Code is available under a CC-BY-4.0 license.

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

## Acknowledgements

Special thanks to Frederique Lisacek and Andrew McDonald for their input on navigating this interdisciplinary topic. Additional thanks to Philip Spahn, Hooman Hefzi, Krystyna Kolodziej and Caressa Robinson for helping edit this manuscript. This work was supported by a Lilly Innovation Fellowship Award (A.R.), the Novo Nordisk Foundation provided to the Technical University of Denmark (NNF10CC1016517, NNF20SA0066621: N.E.L.), NIGMS (R35 GM119850: N.E.L., P41GM103390, P01GM107012, and R01GM130915 to K.W.M.), NICHD (R21 HD080682: L.B.) and USDA (USDA/ARS 6250-6001; M.W.H). This work is a publication of the U.S. Department of Agriculture/Agricultural Research Service, Children's Nutrition Research Center, Department of Pediatrics, Baylor College of Medicine, Houston, Texas. The contents of this publication do not necessarily reflect the views or policies of the U.S. Department of Agriculture, nor does mention of trade names, commercial products, or organizations imply endorsement from the U.S. government.

## Author contributions

B.P.K., A.R., L.B., and N.E.L. designed and performed the study and wrote the manuscript. A.B.B. performed preliminary analysis. A.R. performed modeling analyses. B.P.K. analyzed and interpreted the models analyses. M.A.M. and M.W.H. provided samples. D.C., J.Y.Y., J.A.N., K.W.M., and L.B. performed expression, purification of glycosyltransferases, and kinetic assays. A.R., B.P.K., and N.K. performed literature surveys to determine appropriate candidate genes for each reaction. B.P.K. and B.B. performed motif-level analysis. B.P.K. and A.W.T.C. performed transcription factor analysis.

## Competing interests

The authors declare no competing interests.
