## [Peer Review File · Nature Communications]

Reviewers' Comments:

Reviewer #1:

Remarks to the Author:

In this work Kellman et al. present a combination of glycomics and metabolic modeling to unravel the network underlying biosynthesis of human milk oligosaccharides. The work is high impact because understanding the synthesis of these compounds has important ramifications on the infant gut microbiome.

The method sampled HMO content in nursing mothers and compared the amounts of particular oligosaccharides to mRNA present in the milk from the mammary epithelium. In some cases, enzyme expression (measured by mRNA level) correlated with putative HMO products, but many did not correlate. Thus an approach was used that incorporated the entire possible biosynthetic pathway. The authors determined candidate enzymes and analyzed their potential contributions to biosynthesis via flux balance analysis on an array of possible network structures. The results were confirmed by in vitro assays on substrate specificity of candidate enzymes. In addition, shared transcription factor motifs were identified for various glycosyltransferases, elucidating underlying regulation of these genes.

The systems biology approach used in this work is highly valuable and each part of this work (glycomics, metabolic modeling, kinetic assays, and transcription factor modeling) supports the others. Overall I support it for publication but I would like the authors to address the following concerns:

Major issues

1. Does the model account for the potential pools of nucleotide sugars? Some lack of correlation could be explained by deficiency in these building blocks. Was this addressed?
2. FBA treats all reactions as happening in the same "compartment", e.g. cell. But would there not be heterogeneity in expression of glycosyltransferases in different cells along the epithelium? Is it possible that different cells produce different HMOs? Or that secreted HMOs could be acted upon further down the epithelium (analogous to glycan synthesis?). With only milk as a pooled output, this may be impossible to determine. The authors should address this possibility in the discussion and what might be done to resolve it (e.g., scRNA-seq?).

Minor issues

1. Table 1 refers to colors in the background but I do not see them in the PDF (they are in the .xlsx file). Perhaps this is a formatting issue.
2. Readers not familiar with LiCoRR might find Table 1 difficult to parse given the visual notation used in the figures. Is it possible to add SFNG designs to the table?

Reviewer #2:

Remarks to the Author:

Kellman and colleagues describe a system biology approach to model HMO biosynthesis using transcriptomic and glycoprofilng data supported by kinetic assays. The work addresses an important problem in the construction and implementation of suitable system biology frameworks, which integrate expression data with structural glycan data and computational approaches to model and predict biosynthetic pathways.

The manuscript, supporting materials, statistical models and general code (execution and documentation) is of high quality. The experimental workflows and connections with modelling techniques is well conceived and clearly communicated. Throughout the manuscript the authors provide detailed commentary on strategies used and the generation of reaction networks.

The introduction describes the importance of HMOs and the authors clearly outline biosynthetic rules and machinery involved including the absence of knowledge regarding the precise biosynthetic steps and enzymes involved. Such information is documented in supplementary tables

and suitable references with historical overview of the field is provided. This forms a good basis for those readers not familiar with the field. Importantly, the authors have clearly reviewed the literature to construct an extensive library of reaction rules including the role of multiple isozymes in elongating HMO structures. However, there appears to be an omission of pathway constraints which I have highlighted below.

The work presented is of a collaborative nature that brings together multi-omic platforms with bioinformatic/computational workflows. The work relies on milk samples collected from lactating women (two independent cohorts) between the 1st and 42nd day post-partum that have been used to generate HPLC glycoprofiling and RNA-Seq data etc.

I do have questions and comments:

Major

This reviewer acknowledges the efforts made to clearly explain the reaction models, GT rules and reaction products including the reaction pathways detailed in supplementary. I am concerned that this level of information, especially the reaction pathways Table S2, will be lost on the readers. Do the authors have plans to make this data more accessible to readers e.g. interactive graphics and consider means to denote graphical notations for each of the structures? This is not a criticism, but I feel that this data can be visualised and warrants the effort.

A concern I have is the absence of HPLC data. 2AB labelling of HMOs is mentioned throughout the manuscript and I ask that the authors provide adequate experimental evidence to support the glycan structural assignments. As noted compositional data for cohort 1 was described in a preprint publication in 2019 but cohort 2 is unpublished. I ask that all data and methods are presented including comparison with commercial standards, in part, the proposed models must be validated with experimental data.

For instance, I do question that some HMOs cannot be structurally characterised in Figure 2. The heatmap show concentrations of HMOs and correlations with gene expression, but how does this compare to 2AB LC relative abundance? Please add details on how the absolute and relative concentrations of the 16 most abundant HMOs was measured, and what criteria is used to define the most abundant.

In addition, supporting glycomics data would be useful to validate Figure 4 – the most important reactions – are these abundant structures or biologically important. How realistic are the modelled/predicted structures in this context, do you have suitable data to support the 'numerically' labelled structures?

In the discussion the authors note that FUT3 is selected by default for modelling α 1,4-fucosylation (note incorrect spelling in manuscript) reactions since FUT5 was non-expressed. However, the authors have previously reported that FUT3 exhibits increased 1,3-fucosylation when an acceptor contains an H2 epitope. Here, can the authors explain if this constraint is included in the models and its impact on the generated HMOs. This reviewer was unable to find evidence in the supplementary tables, it is important to discuss how this preference is supported by the reaction networks. Assuming FUT5 is not included can the authors confirm that no fucosylated type 1 structures have been generated? Furthermore, we know that a terminal LacNAc with an α 2,6-sialic acid cannot be fucosylated by FUT3 or FUT5. Although, the authors have clearly described reaction rules can the authors commit on this constraint (and others) and if this is implemented by the network – if so, please add to the discussion with suitable example or link to Table S2. I suggest that known constraints are tabulated and described appropriately.

Supplementary

Section 4.5.1 I suggest rephrasing the Jaccard index and simplifying terminology for general readers. Perhaps using terms such as union and intersection of cohort 1 and 2, essentially overlap between data sets. Some of the language is too complex, in parts, and could be simplified to improve readability and engagement with the audience.

In Section 6.1 the authors mention rules for gene filtering. If possible, can the authors provide an example in support of paragraph 1 (page 36). On the same page it is noted that 25 glycoforms showed no expression on the microarrays but the precise genes are not clearly described. Can the authors alter this paragraph to improve readability of genes absent from the modelling etc. Similarly, the authors mention that FUT5 and FUT9 cannot be ruled out or evaluated, it is not clear if both enzymes were encoded in the reaction rules?

Section 6.2 Can the authors provide suitable cross reference to the 16 most abundant HMOs, and please clarify the statement that 3,929,750 of these subnetworks did not simulate the HPLC data (does this mean that structures predicted were not 'real' i.e. no LC confirmation). By extension of the later question can the authors describe how many/types of structures predicted had no supporting experimental evidence. It is possible that elongated or highly complex structures could not be resolved by the LC conditions?

Section 7.1 should be expanded to provide links to figures supporting the statement that the authors analyzed the promoters and gene expression patterns for common regulatory elements.

Figure S5 needs detailed legend.

Figure S6 and 8 are difficult to read due to its inherent nature but please consider readability.

Figure S12 shows the glycoforms of 5 compositions that cannot be structurally determined. Here, HPLC retention properties should vary for all structures shown and may provide a means to improve the assignments. Can the authors discuss and possibly consider MS analysis of these fractionated HMOs?

Minor

I strongly encourage the authors to check the editing comments left in the supplementary files. For example, Table 1 Excel 'table1_old' rows 18-22 includes a few comments that should be removed.

In Table 1 'samples' are mentioned, however, since this is the first mention of the biological material the authors need to briefly explain the sample sets etc. Additionally, no background colors are included in this table (last sentence; Background colors correspond

Authors encouraged to check formatting, spelling mistakes, and font changes common in supplementary e.g. hexaose, glycosyltransferase (Figure S7).

Different monosaccharide graphical notations are used e.g. S12 and S13.

Random use of bolded text e.g. Glossary in supplementary.

Section 7.2 and 7.3 are not needed or should be included in the introduction and discussion respectively. These sections read as general statements and provide no experimental/data insights.

Reviewer #3:

Remarks to the Author:

Review of Kellman et al. Elucidating Human Milk Oligosaccharide biosynthetic genes through network-based multi omics integration

The work of Kellman et al. aims at elucidating the genes underlying the biosynthesis network of Human Milk Oligosaccharides (HMOs) based on a combination of approaches from constraint-based modelling of metabolic networks (i.e. flux balance analysis, flux variability analysis, and model reduction) with transcriptomics (microarray) and metabolomics data. The work is of interest to the general public since the approach can, in principle, be applied to elucidate networks and

underlying genes in other systems. However, there are numerous clarifications and corrections which need to be considered before assessing the quality of predictions and the new biochemistry / biology learned. This is particularly the case for several discrepancies between the figure captions, methods, and supplementary materials.

Major comments

1. The approach that the authors take uses a set of "source" molecules / compounds (here, glucose) together with a set of (glycosylation) reactions to expand the set of compounds; these are in turn in an iterative process until a set of "target" molecules are reached. That said, it is surprising that no comments to retrosynthesis approaches are made (although it is essentially the reverse process to what the study employs); in addition, the approach is essentially equivalent with that based on substrate-product pairs (which also allows the consideration of information from LC-MS spectra; see <http://www.plantcell.org/content/26/3/929>). Therefore, a more detailed specification of the methodological novelty is needed for audience not familiar with the subject.
2. Line 101 states that absolute and relative concentration of HMOs is measured; it is important to stress how they were measured, what is meant by "relative" (to what?), and, most importantly, how many HMOs are measured already in this part of the manuscript, for clarity.
3. The title to section 2.1 is misleading. No enzyme abundances are measured, so why mention them here?
4. For results shown in Figure 2, the authors should attempt using linear mixed effect models or justify / show why this is not a suitable strategy (it is confusing that GEEs are used, yet, linear trends are shown!) To Figure 2C, the dataset should be changed to cohort. Is there any particular reason why the cohorts / mothers are mixed? This renders it very difficult to assess the variability within mothers, which is an important point for the modelling strategy followed.
5. In caption of Figure 3, the authors write under (F) that FBA was used to estimate the flux through each reaction necessary to simulate the measured oligosaccharide concentrations. However, how concentrations of HMOs are used in the modelling is not specified in the Methods or Supplementary Materials. This renders it difficult / impossible to assess how exactly the metabolomics measurements were used.
6. Line 165 is at odds with the explanation in the caption of Figure 3 C; the text should be consistent. At this point, it is important to indicate if the network forms a directed acyclic graph (i.e. if only irreversible "elementary" reactions are used from the source). Further, the FVA presented in the Supplementary Materials includes only steady-state constraints. Is this the case or additional constraints are imposed (e.g. accumulation of some metabolites must be achieved?) This part should be detailed in the method section, not only the supplementary.
7. Line 169 states that MILP is used to identify subnetworks with minimal number of reactions; however, the constraints to the functionality of the network (e.g. production of some metabolites) is not specified. Hence, this MILP formulation is incomplete. In addition, the MILP shown on pp. 35 of the Supplementary material is not understandable, since the meaning of NZ^k and $NZ^{\{j-1\}}$ are not explained.
8. The biggest issue is that the authors claim that the fluxes are unique for a reduced model. First, the authors need to show that there are NO additional minimal networks, beside the one found. In other words, they have to investigate the space of alternative optima for the MILP (point 7, above) in terms of the binary variables w . Therefore, the calculated scores are based only on a single flux distribution, which –may not be – representative of the functional state of the network.
9. Gene expression is often a poor predictor of flux. The authors should comment on why this is a suitable choice in comparison to, say, proteomics data. In addition, would multivariate regression models based on the isomers not be conceivable in this scenario?
10. What happens if the z-transformation of the model score goes beyond 1.646 (since this threshold does not consider multiple hypothesis testing, which must be included)? Would the cohorts become incomparable, in the sense that the intersection is empty or no meaningful patterns can be detected in the shared reactions? I raise this point, since usually one would aim to reduce the contending models to as few as possible already at this point. This section will have to be reworked with regard to point 8, above.
11. Related to point 10, the Spearman correlation is rather poor, which points that the ordering is close to random. Would one not want to inspect only the agreement in, say, the 10% of best ranked models? This will point at the true quality of the modelling, and will go to say that ordering of the models between the two cohorts becomes irrelevant when worse performing models are

considered.

12. Lines 227 – 229 do not say anything beyond what the data already show, or have I missed anything of importance?

13. Figure 5 is not understandable without digging through the supplementary material. Neither the GLS nor the MSC are defined in the main methods. This has to be resolved to improve readability. In fact, the entire constraint-based modelling section should be pushed to the main text.

14. Related to point 13, it is not clear what is meant by lines 292 – 294 of the supplementary material. What is there are more than one reaction that precedes the reaction with flux f ? The motivation for average over the ratios of fluxes for linkages should also be explained.

15. Title of section 2.4 is misleading, what is meant by “expand our scope”.

16. Section 2.5 comes as an afterthought – I do not see how this is related to the network of generated models. Could the authors investigate and comment if the genes / reactions that are under the control of the same TF show some functional features (e.g. enriched with fully coupled reactions, as indicated in work of *E. coli*, yeast, maize, and *A. thaliana*). This will bring in the network view in this section of the manuscript.

17. The abstract and main text should include a quantified degree of success for the validation. It is currently difficult to assess how many targets were tested, what the hypotheses are, and based on the data, how many of these were accepted / rejected. This is essential to objectively gauge the quality of the modelling work.

Tables

18. The caption of Table 1 mentions background colours, which are not to be seen.

REVIEWER COMMENTS

Reviewer #1 (Remarks to the Author):

In this work Kellman et al. present a combination of glycomics and metabolic modeling to unravel the network underlying biosynthesis of human milk oligosaccharides. The work is high impact because understanding the synthesis of these compounds has important ramifications on the infant gut microbiome.

The method sampled HMO content in nursing mothers and compared the amounts of particular oligosaccharides to mRNA present in the milk from the mammary epithelium. In some cases, enzyme expression (measured by mRNA level) correlated with putative HMO products), but many did not correlate. Thus an approach was used that incorporated the entire possible biosynthetic pathway. The authors determined candidate enzymes and analyzed their potential contributions to biosynthesis via flux balance analysis on an array of possible network structures. The results were confirmed by in vitro assays on substrate specificity of candidate enzymes. In addition, shared transcription factor motifs were identified for various glycosyltransferases, elucidating underlying regulation of these genes.

The systems biology approach used in this work is highly valuable and each part of this work (glycomics, metabolic modeling, kinetic assays, and transcription factor modeling) supports the others. Overall I support it for publication but I would like the authors to address the following concerns:

We thank reviewer #1 for their kind words. It is clear from their summary that they deeply understand the work we present in this manuscript and we look forward to addressing their insightful comments.

Major issues

1. Does the model account for the potential pools of nucleotide sugars? Some lack of correlation could be explained by deficiency in these building blocks. Was this addressed?

Sugar nucleotide pool size may impact glycan synthesis, and are important for modeling the short term dynamics and regulation of glycan synthesis. However, the power of constraint-based modeling methods, which we use here, is that it allows one to analyze the system at a time scale where it can be assumed that the metabolites have settled to steady state concentrations. Thus concentrations of sugar nucleotides are not necessary for the types of analyses here. Indeed, the successful fit to HPLC data showed that the assumption of steady state was reasonable. However, as we have identified a likely topology for the system, one can now do follow-up studies to sample at finer time scales to study the dynamics and regulation of the pathways to see how sugar nucleotides impact the system.

2. FBA treats all reactions as happening in the same “compartment”, e.g. cell. But would there not be heterogeneity in expression of glycosyltransferases in different cells along the

epithelium? Is it possible that different cells produce different HMOs? Or that secreted HMOs could be acted upon further down the epithelium (analagous to glycan synthesis?). With only milk as a pooled output, this may be impossible to determine. The authors should address this possibility in the discussion and what might be done to resolve it (e.g., scRNA-seq?).

Thank you for pointing out these important caveats. Regarding compartmentalization, it is plausible that different HMOs could be produced in different cell types. To study this, one would need to deploy single cell analysis techniques, including scRNA-seq, coupled with glycomic analysis (in particular, single cell glycomics, which remains to be developed), which unfortunately hasn't been done yet. This, however, is an exciting idea for a follow up study, and we mention the caveat and look forward to further studying it.

Fortunately, for this study, the ability for our analysis to reproduce experimental data well (Section 2.2 & 2.3), provides confidence that our models are able to characterize and parametrize the capability of enzymatic activities of the isozymes across the entire tissue rather than describing the kinetics or compartmentalization of individual synthetic steps.

Minor issues

1. Table 1 refers to colors in the background but I do not see them in the PDF (they are in the .xlsx file). Perhaps this is a formatting issue.

We have removed the colors.

2. Readers not familiar with LiCoRR might find Table 1 difficult to parse given the visual notation used in the figures. Is it possible to add SFNG designs to the table?

While SFNG would be our preference, including them in a table is challenging. Instead, we include IUPAC translations next to LiCoRR symbols to help orient the reader. We should note that LiCoRR is less a new representation and more of a description of how Linear Code has been used to describe reaction rules for systems glycobiology models. It, therefore, should be familiar to those in glycan modeling while the IUPAC will ideally help orient those unfamiliar with Linear Code.

Reviewer #2 (Remarks to the Author):

Kellman and colleagues describe a system biology approach to model HMO biosynthesis using transcriptomic and glycoprofilng data supported by kinetic assays. The work addresses an important problem in the construction and implementation of suitable system biology frameworks, which integrate expression data with structural glycan data and computational approaches to model and predict biosynthetic pathways.

The manuscript, supporting materials, statistical models and general code (execution and documentation) is of high quality. The experimental workflows and connections with modelling techniques is well conceived and clearly communicated. Throughout the manuscript the authors provide detailed commentary on strategies used and the generation of reaction networks.

The introduction describes the importance of HMOs and the authors clearly outline biosynthetic rules and machinery involved including the absence of knowledge regarding the precise

biosynthetic steps and enzymes involved. Such information is documented in supplementary tables and suitable references with historical overview of the field is provided. This forms a good basis for those readers not familiar with the field. Importantly, the authors have clearly reviewed the literature to construct an extensive library of reaction rules including the role of multiple isozymes in elongating HMO structures. However, there appears to be an omission of pathway constraints which I have highlighted below.

We appreciate the supportive words from the reviewer and their acknowledgement of the importance of this work. We also appreciate the suggestions on how to strengthen this work, and have addressed each comment below.

The work presented is of a collaborative nature that brings together multi-omic platforms with bioinformatic/computational workflows. The work relies on milk samples collected from lactating women (two independent cohorts) between the 1st and 42nd day post-partum that have been used to generate HPLC glycoprofiling and RNA-Seq data etc.

I do have questions and comments:

Major

This reviewer acknowledges the efforts made to clearly explain the reaction models, GT rules and reaction products including the reaction pathways detailed in supplementary. I am concerned that this level of information, especially the reaction pathways Table S2, will be lost on the readers. Do the authors have plans to make this data more accessible to readers e.g. interactive graphics and consider means to denote graphical notations for each of the structures? This is not a criticism, but I feel that this data can be visualised and warrants the effort.

Thank you very much for this suggestion. The reason that we created a separate supplemental table is to provide interested readers detailed information regarding the reaction rules and model structures. The amount of information is quite daunting and was only intended for people who are interested in replicating the exact models and use them for further bioinformatic analyses. Including the table in the supplement did not impact our ability to support our claims and was intended not to distract the general readers with technical details. As for making the data more accessible to readers, we have now published the data on Zenodo (doi: 10.5281/zenodo.4060217) for enhanced shareability.

Regarding the implementation of an interactive GUI, there are powerful applications (e.g. Glycologue) which allow users to generate useful graphics with our data. It is our intention to compliment the GUI services with accessible, open and reproducible code and data compatible with their tools (e.g. Glycologue networks can be more easily used for genome -> milk sugar predictions using our gene-enzyme mapping). Meanwhile, if we can identify an unmet need within GUI development, we will definitely consider GUI development as we gather more information and assess any additional benefits it might add to the research.

A concern I have is the absence of HPLC data. 2AB labelling of HMOs is mentioned throughout the manuscript and I ask that the authors provide adequate experimental evidence to support the glycan structural assignments. As noted compositional data for cohort 1 was described in a preprint publication in 2019 but cohort 2 is unpublished. I ask that all data and methods are

presented including comparison with commercial standards, in part, the proposed models must be validated with experimental data.

We have now attached the raw HPLC data to the supplement (Table S0 Raw HPLC & Microarray). Since the HPLC experiments for cohort 1 have been described in detail in a previously published paper (McGuire et al), Section 7.2 has been modified to include the reference:

Method 7.2: ...HMO compositions and the absolute abundance measurement for cohort 1 were fully described by McGuire et al. The relative abundance of each glycan in each milk sample is normalized by the total absolute abundance of the 16 most abundant HMO signals (chromatogram signals) for a given sample when used for model construction/fitting, as described by Bao et al¹⁷. Measurements for cohort 2 were previously unpublished but used the same methodology as cohort 1.2)

For instance, I do question that some HMOs cannot be structurally characterised in Figure 2. The heatmap show concentrations of HMOs and correlations with gene expression, but how does this compare to 2AB LC relative abundance? Please add details on how the absolute and relative concentrations of the 16 most abundant HMOs was measured, and what criteria is used to define the most abundant.

In Figure 2, well-known HMO structures were presented next to the heatmap. We agree that “uncharacterized” was too extreme a term. We intended to indicate that the structures were not fully determined though multiple structures have been proposed. We have changed the term from “uncharacterized” to “undetermined.” The exact structures of the ambiguous isomers were shown in Figure S12. Using standards of known concentrations, we were able to obtain the absolute concentrations of HMOs, which allowed us to directly compare the abundances of different HMOs across different samples and time points (Methodology 7.2). Some HMO isomers clearly separate on the HPLC column, e.g. 2’FL vs 3FL or 3’S� vs 6’S� as well as LNFP1 vs LNFP2 vs LNFP3, which means each isomer can be quantified based on standard response curves and reference to the internal standard raffinose. Other HMO isomers, e.g. LSTa vs LSTd elute at the same retention time and cannot be resolved. They have the same mass and the same retention time. MSn fragmentation pattern can distinguish between these isomers, but fragmentation in an isomer mixture loses its quantitative nature, limiting the ability to provide absolute concentrations for these specific isomers. Therefore, some HMO isomers are grouped together.

Finally, Figure 2c does not show HMO-Gene relations but rather HMO concentrations. We have added text to the legend to clarify this. Thank you for pointing out this confusion.

In addition, supporting glycomics data would be useful to validate Figure 4 – the most important reactions – are these abundant structures or biologically important. How realistic are the modelled/predicted structures in this context, do you have suitable data to support the 'numerically' labelled structures?

We thank the reviewer for pointing out this issue. In short, all structures were curated from existing literature focused on solving HMO structures. This information was moved from the introduction to the supplement at some point. We have added specific language to the figure legend where they first appear and to the methods:

Of the many possible HMOs, more than 150 have been identified (Ninonuevo 2006; Wu, 2010; Wu, 2011) and several of the most abundant observed HMOs remain to have ambiguous structures. The natural heterogeneity (branching, isomerization and polarization) of HMO mixture present in milk makes their structural identification and quantitative detection a prohibitive challenge to many current studies (Kobata, 2010; Bode, 2015; Mantovani; 2016). This is due in part to the lack of HMO standards for performing a comprehensive study and structural characterization of each HMO. In this study, we analyzed 16 of the most abundant HMOs, 11 of which have fully determined molecular structure, while the remaining five have multiple alternate candidates structures (listed based on Kobata 2010) We were very careful throughout the paper to distinguish evidence-supported isomeric HMO structures and presenting the possible structures based on known reaction rules (Figure S1,S12,S19).

*-Ninonuevo, M. R. et al. A strategy for annotating the human milk glycome. J. Agric. Food Chem. **54**, 7471–7480 (2006).*

*-Wu, S., Tao, N., German, J. B., Grimm, R. & Lebrilla, C. B. Development of an annotated library of neutral human milk oligosaccharides. J. Proteome Res. **9**, 4138–4151 (2010).*

*-Wu, S., Grimm, R., German, J. B. & Lebrilla, C. B. Annotation and structural analysis of sialylated human milk oligosaccharides. J. Proteome Res. **10**, 856–868 (2011).*

*-Kobata, A. Structures and application of oligosaccharides in human milk. Proc. Jpn. Acad. Ser. B Phys. Biol. Sci. **86**, 731–747 (2010).*

*-Bode, L. The functional biology of human milk oligosaccharides. Early Hum. Dev. **91**, 619–622 (2015).*

*-Mantovani, V., Galeotti, F., Maccari, F. & Volpi, N. Recent advances on separation and characterization of human milk oligosaccharides. Electrophoresis **37**, 1514–1524 (2016).*

In the discussion the authors note that FUT3 is selected by default for modelling α 1,4-fucosylation (note incorrect spelling in manuscript) reactions since FUT5 was non-expressed. However, the authors have previously reported that FUT3 exhibits increased 1,3-fucosylation when an acceptor contains an H2 epitope. Here, can the authors explain if this constraint is included in the models and its impact on the generated HMOs. This reviewer was unable to find evidence in the supplementary tables, it is important to discuss how this preference is supported by the reaction networks. Assuming FUT5 is not included can the authors confirm that no fucosylated type 1 structures have been generated? Furthermore, we know that a terminal LacNAc with an α 2,6-sialic acid cannot be fucosylated by FUT3 or FUT5. Although, the authors have clearly described reaction rules can the authors commit on this constraint (and others) and if this is implemented by the network – if so, please add to the discussion with suitable example or link to Table S2. I suggest that known constraints are tabulated and described appropriately.

We appreciate the reviewer's careful curation of the manuscript and have corrected the typo in this section (Section 3, second paragraph).

Before getting into the details, all reaction constraints are specified in Table 1. All candidate structures for ambiguous structures were listed in figS12.

Regarding the “default” selection, that was “legacy” language from an earlier draft. After adding the RNA-seq post-hoc corroborative check (see supplemental results), we found that FUT5 was expressed in similar milk samples. Therefore, we do not rule it out but rather designate FUT5 as “unevaluated” and FUT3 as “implicated.” We believe this terminology is appropriately couched in the uncertainty afforded by the microarray probe dropout could have led to an inability to measure

FUT5.

The H2 constraint described by the reviewer was not included as a constraint in the model (Table 1). The discussion of previously described GT preferences was used in the discussion to consider the plausibility of our gene-protein-reaction matches.

Regarding the lack of fucosylated type-1 structures, this is not an issue in our model. The summary model (average of the highest performing models) shows no instances where a fucose is added to the same branch as a sialic acid. The selection of “high-scoring” models was made based on the consistency between model behavior and gene expression. The absence of α 1,4-fucose additions to charged structures (despite the fact that we didn’t tell the model to avoid that), provides further evidence that our results are consistent with published literature.

Regarding the terminal sialy-LacNAc, it should be noted that N-glycan terminals are a useful but imperfect model for HMO biosynthesis. Additionally, we do not have in our summary model (fig4) any structures with α 2,6-sialic acid & fucose on the same branch.

Regarding the constraint and preferences of FUT3, we would like to emphasize that each of the reaction rules describes the collective catalytic capability of all the isozymes responsible for that reaction in our models. In this case, based on our scoring metrics and post-hoc validation of our result with expression data, FUT3 was determined to be the major isozymes responsible for both α 1,4-fucosylation and α 1,4-fucosylation; possibly to form Lewis A/B groups type-II fucosylation respectively as discussed in the second paragraph of Section 3. It should be noted that the associations between the model fluxes and FUT3’s expression data were downstream assessments of FUT3’s impacts on the two reactions.

Supplementary

Section 4.5.1 I suggest rephrasing the Jaccard index and simplifying terminology for general readers. Perhaps using terms such as union and intersection of cohort 1 and 2, essentially overlap between data sets. Some of the language is too complex, in parts, and could be simplified to improve readability and engagement with the audience.

We appreciate the reviewer’s suggestion and have simplified the section to improve readability. Supplemental Method 4.5.1: ... *Co-occurrence network analyses involved selection of the top 10% of co-occurring structures; those structures most frequently co-occurring in top-performing models...*

*In Section 6.1 the authors mention rules for gene filtering. If possible, can the authors **provide an example** in support of paragraph1 (page 36). On the same page it is noted that 25 glycogenes showed no expression on the microarrays but the **precises genes are not clearly described**. Can the authors alter this paragraph to improved readability of genes absent from the modelling etc. Similarly, the authors mention that FUT5 and FUT9 cannot be ruled out or evaluated, it is not clear if both enzymes were encoded in the reaction rules?*

Thank you for the suggestion. We have included details for gene filtering decisions, including all expression metrics and removal checks, in the supplemental Table S1 and added the reference in this paragraph. The 25 glycogenes could be readily read from the table under the Removal Check column (“Expression Measured” subcolumn); we have added text to the supplement to more easily direct the reader to this information.

In the case of FUT5 and FUT9, they were not expressed on either microarrays but showed expression in RNASeq and GTex, suggesting possible microarray probe failures. They were *excluded* from calculation since there was no data to use and therefore cannot be confirmed or eliminated as the genes responsible for these reactions. We have clarified this point in the discussion “*therefore FUT5 can neither be evaluated nor dismissed as a candidate gene*” “*FUT9 showed negligible expression in RNA-Seq (3rd Quartile TPM=0.37, Table S 1), yet it is highly expressed (TPM>10) brain and stomach³². Therefore, it is more likely that the distal fucosylation is conducted by another enzyme in vivo while the inner fucosylation is likely performed by either FUT3 or FUT4*”

Lastly, we would like to re-emphasize that the generation of the reaction network is based on literature curated HMO structures and known monosaccharide-additions rather than a *priori* validation of glycosyltransferase isozymes. In other words, GTs were not included in the reaction rules and did not inform network or model construction. We have added clarification in the table 1 legend to avoid this confusion in the future: “*We note that gene “candidates” for each reaction (last column) were not used to inform the metabolic model construction. Candidate genes are those compared to completed metabolic models to evaluate consistency between candidate gene expression and gene-agnostic metabolic model flux through the corresponding reaction.*”

Section 6.2 Can the authors provide suitable cross reference to the 16 most abundant HMOs, and please clarify the statement that 3,929,750 of these subnetworks did not simulate the HPLC data (does this mean that structures predicted were not ‘real’ i.e no LC confirmation). By extension of the later question can the authors describe how many/types of structures predicted had no supporting experimental evidence. It is possible that elongated or highly complex structures could not be resolved by the LC conditions?

I love this question! I actually asked the exact same question early on in this project!

For the clarification of the statement “*The MILP produced 48,914,738 subnetworks. 3,929,750 of these subnetworks were unable to uniquely (upper and lower bound flux were not equal) simulate the HPLC data leaving 44,984,988 candidate models for further examination.*” This means that we removed some models for which there was no unique flux distribution to explain the data for these models. In other words, alternate flux distribution might explain the data equivalently.

Put another way, these “non-unique” models were “underdetermined.” This means there were too many paths through the network to determine the flux. It is not clear that these networks would be informative for determining the plausibility of HMO structures.

Essentially, the issue of flux non-uniqueness is not related to structure plausibility. Flux non-uniqueness is a transitive peculiarity of FBA that is rarely biologically meaningful. Additionally, as previously described, the model did not propose new HMO structures for any of our 16 observed HMOs. All literature-curated candidate structures used in this study were listed in figure S12.

We’ve added citations indicating the sources for structures used to represent these ambiguous structures in introduction paragraph 2, and figure 2.

Section 7.1 should be expanded to provide links to figures supporting the statement that the

authors analyzed the promoters and gene expression patterns for common regulatory elements.
Thank you for the suggestion. We have provided relevant links to the figures on the analyses of promoters and gene expression patterns and several links to the supplement.

Figure S5 needs detailed legend.

Thank you for the suggestion. We have added text in Figure S5 legend so that readers can refer to the main text where the purpose of and the methodology utilizing these gene performance metrics were described in detail.

Figure S : *“Each panel corresponds to one major linkage type and shows the three performance metrics (x-axis) for all considered isoforms (line color) evaluated on data from either cohort 1 or cohort 2 (line type). The y-axis describes the actual value of each metric min-max normalized between zero.. Please refer to Section 2.2 and 7.7 in main text for a detailed description on how these gene performance metrics were used for gene selection”*

Figure S6 and 8 are difficult to read due to its inherent nature but please consider readability.

Thank you for the suggestion. To provide and help to explain the expression data of candidate genes, we have included the expression data in Table S1 - Candidate_Gene_Validation, with which interested readers can easily visualize the data with their own preferences. We also provide the raw data on which these figures were based (Table S0). Finally, we present these figures in high resolution to allow the reader to zoom in. Ultimately, it's challenging to present this amount of visual information cogently. We hope these multiple avenues of accessibility will be sufficient to comfortably inform any reader.

Figure S12 shows the glycoforms of 5 compositions that cannot be structurally determined. Here, HPLC retention properties should vary for all structures shown and may provide a means to improve the assignments. Can the authors discuss and possible consider MS analysis of these fractionated HMOs?

Some HMO isomers clearly separate on the HPLC column, e.g. 2'FL vs 3FL or 3'SL vs 6'SL as well as LNFP1 vs LNFP2 vs LNFP3, which means each isomer can be quantified based on standard response curves and reference to the internal standard raffinose. Other HMO isomers, e.g. LSTa vs LSTd elute at the same retention time and cannot be resolved. They have the same mass and the same retention time. MSn fragmentation pattern can distinguish between these isomers, but fragmentation in an isomer mixture is no longer quantitative, and we cannot provide absolute concentrations for these specific isomers. Therefore, some HMO isomers are grouped together.

Minor

I strongly encourage the authors to check the editing comments left in the supplementary files. For example, Table 1 Excel 'table1_old' rows 18-22 includes a few comments that should be removed.

Thank you for pointing out this issue. We have removed redundant comments from Supplemental Table 1.

In Table 1 'samples' are mentioned, however, since this is the first mention of the biological material the authors need to briefly explain the sample sets etc. Additionally, no background colors are included in this table (last sentence; Background colors correspond

Thank you for the suggestion. We have modified the Table description to clarify the biological samples. We have also removed the reference to background colors.

Authors encouraged to check formatting, spelling mistakes, and font changes common in supplementary e.g. hexaose, glycosyltransferase (Figure S7).

Thank you for the suggestion. We have rechecked the manuscript and the supplement for potential issues about formatting, spelling errors, or unintended font changes.

Different monosaccharide graphical notations are used e.g. S12 and S13.

S13 has been updated with consistent notation.

Random use of bolded text e.g. Glossary in supplementary.

We have double-checked our main text and supplement so that we only used bolded text for captions, important terms, and section titles.

Section 7.2 and 7.3 are not needed or should be included in the introduction and discussion respectively. These sections read as general statements and provides no experimental/data insights.

We agree with the reviewer that Supplemental Discussion S7.2-3 would be best to include in the main text. Unfortunately, out of respect for the journal and their word limits we were forced to move them from the main text. Because these are still important ideas, we chose to retain them in the supplement.

Reviewer #3 (Remarks to the Author):

Review of Kellman et al. Elucidating Human Milk Oligosaccharide biosynthetic genes through network-based multi omics integration

The work of Kellman et al. aims at elucidating the genes underlying the biosynthesis network of Human Milk Oligosaccharides (HMOs) based on a combination of approaches from constraint-based modelling of metabolic networks (i.e. flux balance analysis, flux variability analysis, and model reduction) with transcriptomics (microarray) and metabolomics data. The work is of interest to the general public since the approach can, in principle, be applied to elucidate networks and underlying genes in other systems. However, there are numerous clarifications and corrections which need to be considered before assessing the quality of predictions and the new biochemistry / biology learned. This is particularly the case for several discrepancies between the figure captions, methods, and supplementary materials.

Major comments

1. The approach that the authors take uses a set of “source” molecules / compounds (here, glucose) together with a set of (glycosylation) reactions to expand the set of compounds; these are in turn in an iterative process until a set of “target” molecules are reached. That said, it is surprising that no comments to retrosynthesis approaches are made (although it is essentially the reverse process to what the study employs); in addition, the approach is essentially equivalent with that based on substrate-product pairs (which also allows the consideration of information from LC-MS spectra; see <http://www.plantcell.org/content/26/3/929>). Therefore, a more detailed specification of the methodological novelty is needed for audience not familiar with the subject.

There indeed have been a number of outstanding studies on retrosynthesis, although we took a different approach since it allows us to not only infer the pathways used, but allows us to account for the alternative pathways that could be taken by the glycosyltransferases, since they have act on different oligosaccharides and could in theory produce different HMOs that weren't observed, if the some inaccurate pathways are predicted. However, we have now acknowledged other published retrosynthesis methods, and that deploying variations on such approaches could be invaluable for future studies building upon this work.

Thank you for the interesting comment regarding candidate substrate-product pairs to reconstruct a network of potential biosynthetic routes; this problem is arguably more difficult than ours. Here we have the luxury of some well-defined reactions in an under-defined network that is hierarchical ([10.1016/j.tibs.2020.10.004](https://doi.org/10.1016/j.tibs.2020.10.004)). All of these features permit our simplifying assumptions. Our approach was actually inspired by the work performed by Krambeck, et al. where they used a set of basic rules to reconstruct the protein glycosylation pathways. We have provided the following discussion in the first paragraph of Section 3.

... The biosynthetic model is essentially a probabilistic model where each node represents a specific glycan structure, each edge a known possible enzymatic reaction converting one glycan to another, and edge weight the possibility of such a conversion. This method is highly efficient and intuitive for the iterative HMO biosynthetic network construction due to the modular nature of monosaccharide addition to existing glycan structures during HMO molecular extensions (ref1,3). In comparison with traditional kinetics models of glycan synthesis, the low-parameter framework (ref 1) can utilize either LC or MS data and also allowed inference of enzymatic activities (ref 2) using model parameters, which could be readily validated with the transcriptomics data of involved glycosyltransferases (Supplemental Method 5.3). In comparison with retrosynthesis approaches (ref4), our model could be readily validated by transcriptomics data and the framework allowed further quantitative differentiation of unknown isozyme enzymatic activities (Result Section 2.2, Supplemental Method 6.3)...

1. Spahn, P. N. *et al.* A Markov chain model for N-linked protein glycosylation – towards a low-parameter tool for model-driven glycoengineering. *Metab Eng* **33**, 52–66 (2016).
2. Krambeck, F. J., Bennun, S. V., Andersen, M. R. & Betenbaugh, M. J. Model-based analysis of N-glycosylation in Chinese hamster ovary cells. *PLOS ONE* **12**, e0175376 (2017).
3. Liang, C. *et al.* A Markov model of glycosylation elucidates isozyme specificity and glycosyltransferase interactions for glycoengineering. *Current Research in Biotechnology* **2**, 22–36 (2020).

4. Morreel, K. *et al.* Systematic Structural Characterization of Metabolites in Arabidopsis via Candidate Substrate-Product Pair Networks. *The Plant Cell* **26**, 929–945 (2014).

2. Line 101 states that absolute and relative concentration of HMOs is measured; it is important to stress how they were measured, what is meant by “relative” (to what?), and, most importantly, how many HMOs are measured already in this part of the manuscript, for clarity.

We have now attached the raw HPLC data in Supplemental Table 0. Since the HPLC experiments for cohort 1 have been described in detail in a previously published paper, Section 7.2 has been modified to include the reference and to briefly describe how relative abundances were calculated:

Introduction (<line number here after finalizing manuscript>): ...Absolute and relative concentrations of the 16 most abundant HMOs were measured (Method 7.2)...

*Method 7.2: HMO compositions and the absolute abundance measurement for cohort 1 were fully described by PMID: 33328537. The relative abundance of each glycan in each milk sample is normalized by the total absolute abundance of the 16 most abundant (typically >95% of HMO mass per sample(PMID: 33328537)) HMO signals (chromatogram signals) for a given sample when used for model construction/fitting, as described by Bao *et al*¹⁷. Measurements for cohort 2 were previously unpublished but used the same methodology as cohort 1.*

3. The title to section 2.1 is misleading. No enzyme abundances are measured, so why mention them here?

The correlations between related glycosyltransferase expressions and the HMO concentrations which require the enzyme were calculated and presented in Figure 2. We have revised the title to be more clear. To emphasize this point, we have modified a portion of Section 2.1:

...But, examining only subjects with functional FUT2 (Secretors), we found FUT2 expression levels and the concentration (nmol/ml) of HMOs containing α -1,2-fucosylation do not correlate in sample-matched microarray and absolute HMO abundance measurements by HPLC (Method 7.2, Figure 2)...

4. For results shown in Figure 2, the authors should attempt using linear mixed effect models or justify / show why this is not a suitable strategy (it is confusing that GEEs are used, yet, linear trends are shown!) To Figure2C, the dataset should be changed to cohort. Is there any particular reason why the cohorts / mothers are mixed? This renders it very difficult to assess the variability within mothers, which is an important point for the modelling strategy followed.

Thank you for asking about mixed effect models (nobody ever asks about this)! Through previous experience ([10.1136/gutjnl-2016-312819](https://doi.org/10.1136/gutjnl-2016-312819)) we have found the mixed effect models struggle to fit HMO concentration responses. GEE, on the other hand, likely because of their quasi-likelihood function and subsequent ambivalence with regards to the exact response distribution, are more likely to stably converge when describing HMO trends from repeated measures.

The linear trends in **Figure 2A/2B** were drawn to illustrate the poor correlations between the FUT2 expressions and related HMOs. Though GEE are certainly more complex instruments, they can

be described by analogy to a weighted aggregation of regressions within each subject (or repeat group). Therefore, we believe a trendline for each subject was an appropriate illustration of a trend measured using GEE. We have modified Section 2.1 and the description of Figure 2 to clarify the confusion:

Section 2.1, paragraph 1: ...*Generalized Estimating Equations (GEE) were used to measure the correlations and showed no significant positive association (2nd FL Wald $p = 0.056$; LNFI Wald $p = 0.34$).*

Figure 2: ... *The linear trends were drawn to visually and qualitatively demonstrate the poor correlations...*

We also appreciate your suggestion regarding Figure 2C and have modified Figure 2C so that mothers from the same cohorts were clustered together.

1. Lenz, S. T. Alan Agresti (2013): Categorical data analysis. Stat Papers 57, 849–850 (2016).
2. Nelder, J. A. & Wedderburn, R. W. M. Generalized Linear Models. Journal of the Royal Statistical Society: Series A (General) 135, 370–384 (1972).

Regarding figure 2C, we presented the data clustered to highlight the comparability across cohorts while displaying as much data as possible. We could change this to subject/time sorted but our preference is to highlight the comparability across cohorts.

5. In caption of Figure 3, the authors write under (F) that FBA was used to estimate the flux through each reaction necessary to simulate the measured oligosaccharide concentrations. However, how concentrations of HMOs are used in the modelling is not specified in the Methods or Supplementary Materials. This renders it difficult / impossible to assess how exactly the metabolomics measurements were used.

Thanks for highlighting this. We have added a description on how we used the concentrations of HMOs to compute the flux distribution in the supplementary section 5.4 :

“Flux Balance Analysis (FBA) calculates a unique flux distribution (i.e., metabolic fluxes through all reactions in a network) using Linear Programming (LP) (i.e., minimization or maximization of one specific reaction of a network - equation 1) while Flux Variability Analysis performs a min-max LP problem (i.e. each reaction of the network is maximized and subsequently minimized - equation 2). FBA calculates the flux vector that promotes a specific reaction within a network (described in equation 3) while satisfying a set of constraints (defined in equation 4). While the optimal value for a defined v_i is always unique, the associated flux distribution v is usually not unique, as the LP problem is underdetermined. Thus, FVA can be used to evaluate the minimum and maximum range of each reaction flux that can still satisfy the systems of linear equations and associated constraints (equations 3-4)⁹⁻¹¹.

$$\text{Min or Max } v_i \tag{1}$$

$$v_{i,lower} / v_{i,upper} = \text{Min} / \text{Max } v_i \quad \forall v_i, i = 1, \dots, N \tag{2}$$

Subject to

$$Sv = 0 \quad (3)$$

$$lb \leq v \leq ub \quad (4)$$

where v is the vector of specific reaction rates (i.e. metabolic fluxes), N is the number of fluxes in v , lb and ub are respectively the constraints set on the lower and upper bounds for the individual flux values, $v_{i,upper}$ and $v_{i,lower}$ are respectively the feasible upper and lower values of each flux v_i satisfying the system of linear equations (equations 3-4).

To trim the Complete Network using FVA, the lower and upper bounds constraints of all reactions leading to the secretion of HMOs were set to zero except for the ones leading to the 16 experimentally-measured HMOs which were set to a value of 1/16 in order to account for their synthesis. To compute the flux distribution necessary to synthesize the HMOs up to the measured concentrations using FBA, the lower and upper constraints bounds (lb and ub) of the 16 sink reactions corresponding to each measured HMOs were set to a value corresponding to the relative concentration in HMOs observed (concentration of the specific HMO divided by the total amount of HMOs). FBA is solved using the function `optimizeCbModel` and FVA using the function `fastFVA` both available in the `CobraToolbox 3.012`.

6. Line 165 is at odds with the explanation in the caption of Figure 3 C; the text should be consistent. At this point, it is important to indicate if the network forms a directed acyclic graph (i.e. if only irreversible “elementary” reactions are used from the source). Further, the FVA presented in the Supplementary Materials includes only steady-state constraints. Is this the case or additional constraints are imposed (e.g. accumulation of some metabolites must be achieved?) This part should be detailed in the method section, not only the supplementary.

We made major modification to the description of the usage of FVA for network trimming in the Supplementary section 5.4 (see response to your comment 5 above) and hope it will better clarify the network structure and the methodology. Due to the large numbers of publications on the established FVA methodology, we included the technical details in Supplemental Method 5.4 for more modeling-oriented readers so that the workflow can be better presented to the general audience.

7. Line 169 states that MILP is used to identify subnetworks with minimal number of reactions; however, the constraints to the functionality of the network (e.g. production of some metabolites is not specified. Hence, this MILP formulation is incomplete. In addition, the MILP shown on pp. 35 of the Supplementary material is not understandable, since the meaning of NZ^k and NZ^{j-1} are not explained.

We have modified the description of the algorithm as follows to address the comment and included the details in Supplemental Method 5.5:

The enumeration of all alternate minimal reaction sets solving equally an LP such as $Sv = 0$ can be formulated with the following recursive MILP problem^{108,109}:

$$\text{Min } Z = \sum_{i=1}^N w_i \quad (1)$$

Subject to

$$Sv = 0 \quad (2)$$

$$\sum_{i \in NZ^{K-1}} y_i \geq 1 \quad \text{with } y_i \in \{0,1\} \quad (3)$$

$$\sum_{i \in NZ^k} w_i \leq |NZ^k| - 1 \quad \text{with } w_i \in \{0,1\} \text{ and } k = 1:K - 1 \quad (4)$$

$$w_i + y_i \leq 1 \quad \forall i \quad (5)$$

$$w_i \cdot lb \leq v_i \leq w_i \cdot ub \quad \forall i \quad (6)$$

Where S is the stoichiometric matrix, v is the vector of specific reaction rates (i.e. metabolic fluxes), N is the number of fluxes in v , y and w are vector of binary variables and lb and ub are respectively the lower and upper bounds for the individual flux values. The lower and upper bounds are respectively set to 0 and 1 for all the reactions in the network except for the 16 sink reactions corresponding to each measured HMOs which are constrained to a value of 1/16 in order to account for their synthesis.

The set of constraints (3-6) are used for changing the basis and identifying a new extreme point corresponding to one of the alternate optima. Specifically, at each iteration, K , at least one of the non-zero fluxes from the previous solution (NZ^{K-1}) must be set to zero, where the binary variable y_i is 1 if that flux is selected to be removed from the basis at iteration K (equation 3). The binary variable w_i is subsequently forced to zero if y_i is one (equation 5), and the upper and lower bounds for that particular flux are then constrained to zero (equation 6). Equation 4 ensures that alternate bases are not revisited by eliminating at least one non-zero variable found in previous iterations. Typically, the search algorithm stops when no other solution with the same objective function can be found in the MILP. In the context of this work, we slightly modified this end criterion in order to investigate all minimal networks: the search stops when the generated networks no longer present a unique flux distribution using FVA. The MILP problem formulated as above is solved using the function `solveCobraMILP` in the `CobraToolbox 3.0`.

8. The biggest issue is that the authors claim that the fluxes are unique for a reduced model. First, the authors need to show that there are NO additional minimal networks, beside the one found. In other words, they have to investigate the space of alternative optima for the MILP (point 7, above) in terms of the binary variables w . Therefore, the calculated scores are based only on a single flux distribution, which –may not be – representative of the functional state of the network.

The MILP formulation has been clarified to describe how we investigated the entire space of alternate minimal networks and ensure that all generated networks are associated with a unique flux distribution (see response to comment 7 above or Supplemental Method 5.5).

9. Gene expression is often a poor predictor of flux. The authors should comment on why this is a suitable choice in comparison to, say, proteomics data. In addition, would multivariate regression models based on the isomers not be conceivable in this scenario?

We agree with the reviewer that proteomic data should be a better predictor of fluxes than gene expression data due to uncertainty in turnover from mRNA to protein. Unfortunately, there are major technical hurdles in measuring and quantifying the proteome within mammary epithelial cells given the large amounts of protein secreted into the milk, which would mask the small amounts of protein from cells that have been shed. mRNA, however, is easier to detect and separate from secreted proteins. Also, while absolute mRNA expression doesn't correlate well with absolute flux, especially for central carbon metabolism, and other pathways with extensive metabolic regulation (<https://www.frontiersin.org/articles/10.3389/fpls.2014.00668/full>). However, relative fold changes in mRNA abundance does often correlate well with changes in flux for pathways with limited enzyme regulation, as we have shown for microbes (Lewis, Mol Syst Bio, 2010; Nam, Science, 2012).

Also, there are no perfect models and the presence of an enzyme does not guarantee its action. The action of an enzyme is contingent on the concentration of substrates and products and the local microenvironments. We were excited to work with the only HMO & gene expression sample matched data in the world at the time, and did so cautiously knowing that gene expression is necessary for enzyme action but far from sufficient. This is exactly the reason that gene expression cannot predict HMO concentration; as visually described in Fig2A-B, the corresponding text, and the unsuccessful regression models. We believe a modeling approach was necessary to account for the variation in upstream limitations and that our restrained use of gene expression data (despite its imperfections) was appropriate. We discuss proxy-based glycan biosynthesis models at length in a recent review: <https://doi.org/10.1016/j.tibs.2020.10.004>

More specifically, we used gene expression as a proxy to inform us of the global dynamic of the data. The concentration data used for flux simulation, together with the measured variation of gene expression, can inform us how may "support" flux when compared across different samples and groups. Indeed, transcriptomics data has been widely adopted by the FBA-based metabolic modeling framework to constrain model parameters and create tissue-specific models (ref 1-4). Since each glycosyltransferase broadly participates in many reactions during HMO synthesis, such dynamics can be more salient and strictly investigated in our study (Result 2.2, 2.3, Supplement Method 5.3, 6.3). Most importantly, we note that the use of transcriptomics with the model allowed us to predict which enzymes could be involved, and we note that these are predictions, but went on to do enzyme assays to validate a subset of the newly predicted enzymes.

1. Heirendt, L. *et al.* Creation and analysis of biochemical constraint-based models: the COBRA Toolbox v3.0. *Nat Protoc* **14**, 639–702 (2019).
2. Schellenberger, J. *et al.* Quantitative prediction of cellular metabolism with constraint-based models: the COBRA Toolbox v2.0. *Nat Protoc* **6**, 1290–1307 (2011).
3. Kim, M. K. & Lun, D. S. Methods for integration of transcriptomic data in genome-scale metabolic models. *Computational and Structural Biotechnology Journal* **11**, 59–65 (2014).
4. Gowen, C. M. & Fong, S. S. Genome-scale metabolic model integrated with RNAseq data to identify metabolic states of *Clostridium thermocellum*. *Biotechnology Journal* **5**, 759–767 (2010).

10. What happens if the z-transformation of the model score goes beyond 1.646 (since this threshold does not consider multiple hypothesis testing, which must be included)? Would the cohorts become incomparable, in the sense that the intersection is empty or no meaningful

patterns can be detected in the shared reactions? I raise this point, since usually one would aim to reduce the contending models to as few as possible already at this point. This section will have to be reworked with regard to point 8, above.

Thank you for asking about this! I hesitated to include comments on the multiple test correction since it's not relevant but it is confusing (as demonstrated by your concern). A z-score normalization is simply performing the z-score transformation to leverage some of the convenient elements of the distribution. This is different from a z-test where a z-score is associated with a p-value for the purposes of establishing significance. We did not perform multiple statistical tests (or even one) at this stage in the calculation, therefore there is no need for a multiple test correction. Had we instead declared the top 5% of models "true" a multiple test correction would have been necessary. Instead, we refer to them as high-scoring (effectively high ranking) on the assertion that even if no single model was true, they should be enriched for true features. This ensemble approach was useful due to the heterogeneity of the underlying biology and the indeterminate existence of a single "true" HMO biosynthesis model.

As we have explained in our responses to comment 7 and 8, we have clarified how our algorithm explored the entire space of minimal and ensured that all generated networks are associated with unique flux distributions. This bottom-up approach allowed further examinations of the top-ranked models using gene-linkage score model score contribution, and the importance of reduced networks (Supplemental Method 4.4,4.5), relative to the background of lower-ranked models. Reactions in high-ranking models (those selected by cohort 1, cohort 2, or both) were highly enriched in high-ranking models relative to lower-ranking models hypergeometric enrichment (Method 7.5).

11. Related to point 10, the Spearman correlation is rather poor, which points that the ordering is close to random. Would one not want to inspect only the agreement in, say, the 10% of best ranked models? This will point at the true quality of the modelling, and will go to say that ordering of the models between the two cohorts becomes irrelevant when worse performing models are considered.

We considered a similar approach to the one you describe (using the consensus of models chosen by the two datasets). Ultimately, we decided that would neutralize the potential use of these independent cohorts for validation. You likely noticed that throughout the modeling effort (Fig3-5) cross-comparison between cohort1 and cohort2 was used to orient the reader to the consistency between these models (this is done to "*point at the true quality of the modelling*" as you say). Yet, we do not combine information across cohort 1 and cohort 2 derived models until Fig5 when we combine performance metrics (GLS, MSC, PROP) for each reaction-linkage pair across cohorts. This was a deliberate separation to leverage the independence of these datasets for validation purposes.

12. Lines 227 – 229 do not say anything beyond what the data already show, or have I missed anything of importance?

Thank you for pointing this out. These lines are a commentary on a nuanced comment from a colleague regarding whether the enrichment of type-1 structures was artifactual. They do not belong in the main text and have been removed.

13. *Figure 5 is not understandable without digging through the supplementary material. Neither the GLS nor the MSC are defined in the main methods. This has to be resolved to improve readability. In fact, the entire constraint-based modelling section should be pushed to the main text.*

Thank you for pointing out this miscommunication. We have added additional text to the fig5 legend to help redirect readers to the precise definitions already present in the main and supplemental text to avoid this confusion in the future. We agree that GLS and MSC could be briefly introduced here to inform the readers. We have modified the figure description so that interested readers can be guided to the corresponding Supplemental sections for more details:

Figure 5: *...Briefly, the GLS measures the correlation of different glycosyltransferase isozyme expression and the simulated fluxes of HMO biosynthesis, whereas the MSC was an enrichment metric assessing the impact of glycosyltransferase isozyme expression on model scores (Methods 7.7, S4.4, S4.5.2, S5.3)...*

While we would love to include as much detail as possible in our approach to identify the candidate isozyme genes, we were concerned that a large number of technical details might distract the general readers (especially glycobioologists) from understanding at a higher level how we determined gene importance.

We have now defined GLS and MSC in the text and figure caption, and also note that they are post-modeling comparisons between flux and gene expression. They are not involved in the flux modeling.

14. *Related to point 13, it is not clear what is meant by lines 292 – 294 of the supplementary material. What is the importance of more than one reaction that precedes the reaction with flux f? The motivation for average over the ratios of fluxes for linkages should also be explained.*

Thank you for asking about this. The flux normalization was one of my favorite solutions in this project! In comment #9, you pointed out that gene expression is an imperfect predictor of enzyme activity; we point out that proteomics is imperfect too (though better). Both are imperfect because they don't consider the environment! To mitigate this issue, we normalized reaction flux (r_1) by the flux of reactions (r_a, r_b, r_c) that synthesize the substrate for r_1 . That way, instead of comparing expression directly to flux, we compared it to the flux/precursor_flux ratio; effectively, substrate limited flux. Though we know that flux includes a cognizance of precursor limitation, we were surprised to find that expression-flux correlation was much lower and less significant than expression-normalized flux correlation (figS11). Ultimately we decided to move forward with the latter.

We have emphasized the importance of reasoning for prior-normalized flux and the provided brief reasoning of GLS/MS as follows:

Figure S 11 - Comparison of total flux and normalized flux. *...In this method we used prior-normalized flux to control for the availability of the precursor...*

Supplemental Method 4.4 ...Doing so, every subnetwork is associated with 10 normalized flux values for each HMO measurement. The normalized flux values represent the activity levels of the 10 types of transferase reactions required for the biosynthesis of measured HMOs ...

Supplemental Method 4.5.2

...The **gene-linkage score metric** (GLS), described in section S1.1.4, is the Spearman correlation between gene expression and normalized flux: $\rho_{gl} = cor(E(g) \sim \underline{f}_{L_r \omega D_m})$. This metric provides information on every gene in every model regardless of the relevance of the gene to the model. This helps identify genes with low congruence because every gene is evaluated.

The **model score contribution** (MSC) is the Pearson correlation between the model score and the gene-linkage score for a specific gene, $cor(S_\omega \sim \rho_{gl})$. This is a more continuous approach to the question posed by the enrichment metric. This is a high granularity metric that can distinguish genes that highly influence the overall score.

15. Title of section 2.4 is misleading, what is meant by “expand our scope”.

Thank you for your suggestion. We have modified the section title to *2.4 Kinetic assays to confirm the gene-reaction associations*. We believe this new title can better summarize Section 2.4 without ambiguity.

16. Section 2.5 comes as an afterthought – I do not see how this is related to the network of generated models. Could the authors investigate and comment if the genes / reactions that are under the control of the same TF show some functional features (e.g. enriched with fully coupled reactions, as indicated in work of E. coli, yeast, maize, and A. thaliana). This will bring in the network view in this section of the manuscript.

Section 2.5 was brought in after the primary analysis. Despite not being part of the original plan, we believe the analysis positively contributes to the paper. Essentially, we wondered if there was some additional evidence we could gather that would (1) corroborate the co-activity of the genes we selected and (2) motivate further exploration; we decided to perform a 2-method TF-motif analysis. There is certainly more to do here but since this was presented as a corroborative analysis.

Because we aim to provide the best analysis appropriate for the paper, we appreciate the reviewer's suggestion. Unfortunately, due to the extreme divergence of primate glycosylation (even mouse milk oligosaccharides provide surprisingly limited information in the study of human milk oligosaccharides) and due to the fact that bacterial and plants do not produce milk oligosaccharides, glycosylation within these model organisms is unlikely to provide relevant information to this study.

To bolster the validity of our analysis, we used two orthogonal approaches to TF-discovery and reported the commonly discovered motifs.

In Section 2.5 and supplemental section 7.1, we have provided several examples of co-regulation of glycogenes associated with specific promoter and TF motifs:

...SP1, EGR1, ETS1, ETV4 and ERG are all predicted to positively influence expression associated with the biosynthetically related HMOs: 3'SL, 3FL, LSTb and DSLNT; 3'SL and 3FL share a common substrate (lactose) while LSTb is a likely precursor to DSLNT. The motif-level analysis showed opposing regulation between IKZF1: upregulating gene expression signatures associated with the 3'SL and LSTb substructure abundance¹⁷ (X34 and X62 respectively, see **Figure S 19**) and downregulating gene expression associated with GlcNAC-lactose, LNT and LNFPI substructure abundance (X18, X40 and X65 respectively, see **Figure S 19**), while EGR1, ERG and ETS1 have the opposite predicted impact (**Figure S 17**). The motif-level predictions are consistent with the HMO-level predictions of upregulation on 3'SL and LSTb while adding an additional point of contrast. While EGR1, ERG and ETS1 are predicted to increase production of sialylated HMOs, they may have the opposite impact on LNFPI...

These analyses were relevant to our study because they not only lend credence to our gene-expression analyses by showing structural co-regulatory coordination but also leveraged additional expression information beyond glycomenes.

17. The abstract and main text should include a quantified degree of success for the validation. It is currently difficult to assess how many targets were tested, what the hypotheses are, and based on the data, how many of these were accepted / rejected. This is essential to objectively gauge the quality of the modelling work.

We thank the reviewer for this important comment. We realize we didn't have an RMSE or AUC to anchor our model performance but rather focused on consistency with the literature. Due to the small number of known/confirmed reactions, we were concerned such statistics could be misleading. We have added some text to emphasize our numerical performance:

Abstract: *Our model aggregation approach recovered 2 of 2 previously known gene-enzyme relations and 2 of 3 empirically confirmed gene-enzyme relations. The top genes proposed for the remaining 5 linkage reactions were consistent with previously published literature.*

Discussion: *Compared to well-known (i.e. FUT2 & ST6GAL1) events and empirically validated (confirmed by kinetics assays and expressed in milk) we observed 4 true-positive, 7 true-negative, 1 false-positive and 1 false-negative gene-enzyme prediction using our approach (sensitivity=0.875, specificity=0.8, PPV =0.875, sensitivity=0.875, specificity=0.875). Our approach correctly resolved both well-known reactions. Kinetics assays showed our approach selected milk-expressed substrate accepting gene-enzyme pairs for reactions L1 and L4 but not L10. Kinetics assays also found a false-negative prediction for reaction L4 and a false-positive prediction for reaction L10.*

Tables

18. The caption of Table 1 mentions background colours, which are not to be seen.

We have removed the colored backgrounds.

Reviewers' Comments:

Reviewer #1:

Remarks to the Author:

The authors have satisfied my concerns and I recommend for publication.

Reviewer #2:

Remarks to the Author:

This reviewer acknowledges the excellent responses made by the authors, which addresses most of the question raised. These responses have been presented and justified well and suitable changes to the manuscript/supplementary information have been made.

However, I still have a major concern with the lack of chromatographic evidence presented in the manuscript. The authors mention that "Measurements for cohort 2 were previously unpublished but used the same methodology as cohort 1". For completeness it is strongly recommended that this data is provided since it forms a vital component of the conclusions made and models built etc. Analytical readers will be keen to see LC profiles. Although, the authors have added Table S0 this is really very difficult to interpret and the absence of 2AB profiles leaves the manuscript potentially open to critical comments on glycan characterisation etc.

Overall, this is excellent work which is well presented.

Reviewer #3:

Remarks to the Author:

The revised version is more technically sound as a result addressing some of the comments in the previous round of reviews (particularly the MILP procedure for the generation of reduced models). There are, however, few important remaining points that are still unclear and not properly addressed:

1. The biggest issue is again with the integration of relative metabolomics data (point 5 in my first review). First the author state that all sink reactions to the measured HMOs are of fixed fluxes 1/16 (meaning that all HMOs are of same relative concentration); however, in the next sentence of the Supplementary material, these sink fluxes are allowed to have lower and upper bounds determined from the relative abundance. The first overrides the second constraints, and therefore, the relative concentrations are not properly integrated. The authors must ensure that even when the lower and upper bounds of these sink reactions are fixed from experiments, the simulated flux distribution must ensure that the sum of the fluxes of sink reactions is 1. This needs careful reconsideration and fixing along with the other points below.
2. Point 10 in my previous review, regarding the z-score for model ranking remains unanswered; the current threshold is arbitrarily imposed and robustness of the findings to other values for the cut-offs would further justify and strengthen the findings.
3. Point 14 in my previous review, Figure S11 does not suffice for the explanation, as it does not show how large the values of the Spearman correlation coefficient are (only the p-values are shown). In addition, the selection of which normalization is included in the manuscript is done a posteriori (one p-values are determined), and I still do not see a mechanistic justification for it.
4. Point 16 in my previous review aimed to direct the authors to inspect full coupling of reactions that are downstream of same TF regulator (shown as a principle of network organization in other organisms mentioned in my review!); this will be a nice complementation to the presented results from the TF motif search.

Minor points

5. The supplementary file contains comments from the co-authors that should be removed.

REVIEWER COMMENTS

Reviewer #1 (Remarks to the Author):

The authors have satisfied my concerns and I recommend for publication.

We thank reviewer #1 for their thoughtful comments and insight. Thank you for helping us improve the manuscript and thank you for supporting its publication.

Reviewer #2 (Remarks to the Author):

This reviewer acknowledges the excellent responses made by the authors, which addresses most of the question raised. These responses have been presented and justified well and suitable changes to the manuscript/supplementary information have been made.

We thank reviewer #2 for their thoughtful comments and insight. Thank you for your kind words, important questions, and support for the publication.

However, I still have a major concern with the lack of chromatographic evidence presented in the manuscript. The authors mention that "Measurements for cohort 2 were previously unpublished but used the same methodology as cohort 1". For completeness it is strongly recommended that this data is provided since it forms a vital component of the conclusions made and models built etc. Analytical readers will be keen to see LC profiles. Although, the authors have added Table S0 this is really very difficult to interpret and the absence of 2AB profiles leaves the manuscript potentially open to critical comments on glycan characterization etc.

On the advice of our empirical collaborator, we've included an example chromatogram to demonstrate how the data was analyzed (fig S22). We hope this is sufficient to address the reviewers concerns.

Overall, this is excellent work which is well presented.

Reviewer #3 (Remarks to the Author):

The revised version is more technically sound as a result addressing some of the comments in the previous round of reviews (particularly the MILP procedure for the generation of reduced models).

There are, however, few important remaining points that are still unclear and not properly addressed:

1. The biggest issue is again with the integration of relative metabolomics data (point 5 in my first review). First the author state that all sink reactions to the measured HMOs are of fixed fluxes 1/16 (meaning that all HMOs are of same relative concentration); however, in the next sentence of the Supplementary material, these sink fluxes are allowed to have lower and upper bounds determined from the relative abundance. The first overrides the second constraints, and therefore, the relative concentrations are not properly integrated. The authors must ensure that

even when the lower and upper bounds of these sink reactions are fixed from experiments, the simulated flux distribution must ensure that the sum of the fluxes of sink reactions is 1. This needs careful reconsideration and fixing along with the other points below.

We would like to thank the reviewer for highlighting this and we agree with the reviewer that, as written, the explanation provided in the section 5.4 of the supplementary materials might lead to a misunderstanding of the reader. Actually, the two sets of constraints described by the reviewers are not applied at the same step of the procedure:

- for procedure step of the complete network trimming (Figure 3C), we used FVA and constrained all sink reactions to the measured HMOs to fixed fluxes of 1/16. The algorithm and associated constraints were only used to generate the reduced network of the HMO biosynthesis pathway (Figure 3D).
- for procedure step of the generation of the “Model Flux” (Figure 3F), we used FBA and allowed the sink fluxes associated to the measured HMOs to have lower and upper bounds determined from relative abundance. Note that we can ensure that the sum of sink fluxes is equal to 1 because the upper and lower bounds of each sink reaction are both equal to the measured relative abundance of the associated HMO.

We re-wrote the last paragraph of the section 5.4 of the supplementary material to ensure the clarity of this explanation.

2. Point 10 in my previous review, regarding the z-score for model ranking remains unanswered; the current threshold is arbitrarily imposed and robustness of the findings to other values for the cut-offs would further justify and strengthen the findings.

We apologize for the confusion and thank you for highlighting this important point. As we previously explained, model ranking was based on a z-score normalization, not a z-test. A z-score normalization has some useful qualities, but it is importantly not a statistical test. Therefore, thresholding, cutoffs or multiple test corrections are not relevant at this stage. As is common practice in discretized ranking tests, we set a heuristic threshold and compared model features that were higher (above the threshold) and lower (below the threshold). Our method is similar to a hypergeometric enrichment performed in differential expression. One performs differential expression, asserts a heuristic cutoff based on LogFC, FDR and base-mean (the exact cutoff is rarely important as long as you approximately segregate relevant from irrelevant, you will recover the relevant pathways), then compares the more interesting genes to the less interesting genes to determine which interesting things are over-represented.

Over-representation of “more interesting” model features in the set of better (not best) performing models is what we were exploring. That is why we do not have strong justification for the distinction between better and worse models. Instead, we have strong justification and multiple test corrections for the actual statistic: hypergeometric enrichment of “more interesting” model features in the set of better performing models.

Because of the importance of the distinction between normalization and statistical testing and the potential damage conflation could do to the manuscript’s credibility, we have added a note to the supplemental methods (section S4.4.2) explaining our choice.

Thank you for highlighting the importance of this potential miscommunication.

3. Point 14 in my previous review, Figure S11 does not suffice for the explanation, as it does not show how large the values of the Spearman correlation coefficient are (only the p-values are shown). In addition, the selection of which normalization is included in the manuscript is done a posteriori (one p-values are determined), and I still do not see a mechanistic justification for it.

Thank you for pointing this out. We have added an additional redirection to the figure caption “See Section S4.4 for additional information” for interested readers. In tandem with the supplemental glossary, this should be sufficient to redirect interested readers.

Because p-values are perfectly determined by correlation and sample size, it is redundant to show both correlation and p-value for a fixed sample size (as is the case in FigS11).

As we mention in our previous response, we do not rely on FigS11 as a complete explanation of the normalization but rather redirect to section S4.4 which describes the method in detail. We choose to normalize flux by precursor flux to refocus the predicted flux in the context of its precursors. Of course, a metabolic network will consider multiple such precursor dependencies. But, in our application, we are most interested in the direct substrate-precursor/substrate/product relations. Additionally, a similar precursor normalization method has been shown to be effective in previous publications for general glycosylation modeling (10.1016/j.ymben.2015.10.007, 10.1002/biot.201600489). We have added text to the Supplement citing these methods: *“The precursor normalization applied here is similar to prior published work^{9,10}. In previous work, the normalization was used to convert flux values to probabilities and focus the computation on local inter-dependencies. Here we use precursor normalization to focus our analysis on local precursor dependencies. We observe an increase in gene-linkage score (FigS11) as evidence of the improved consistency between gene expression and flux.”*

Regarding the timing of the p-value calculation, we don’t believe this is changeable. The p-values describing a correlation cannot be calculated before the correlated values are generated. If the concern is that the normalization was performed correlations were calculated, that is partially true. When exploring the best method of normalizing flux for the purpose of flux-expression comparisons, we compared the p-value distributions. Finding that the normalized flux produced more robust results (FigS11), we chose to proceed with that method which had been previously described and vetted in other published work.

We also note that results were followed up with empirical enzyme kinetics validation, thus providing confidence in the validity of these earlier analyses.

4. Point 16 in my previous review aimed to direct the authors to inspect full coupling of reactions that are downstream of same TF regulator (shown as a principle of network organization in other organisms mentioned in my review!); this will be a nice complementation to the presented results from the TF motif search.

I love this idea! Thank you for clarifying! I do not think we can do this important work justice as a supplemental add-on. We added two independent TF analyses (motif-based with MEME and expression based with IPA) as a prospective example of how the gene-reaction assignments

(the focus of the paper) could be applied beyond the constraints of the manuscript. Here, we provide a robust initial look at TF-gene predictions. To prune those initial predictions into a set of high-confidence results will require an additional paper that will certainly include the coupling reaction analysis you suggested.

Minor points

5. The supplementary file contains comments from the co-authors that should be removed.

Resolved

Reviewers' Comments:

Reviewer #2:

Remarks to the Author:

The authors have addressed all points raised. Thank you.

Reviewer #3:

Remarks to the Author:

The authors have provided detailed replies in attempt to address most of the raised points in the last round of reviews.

Two issues remain:

To point 2 in the previous round of reviews, the authors do not show how robust the findings (from enrichment analyses) are to changing the z-score cut-off. This can be simply done by checking which of the "more interesting" model features remain as such after taking a higher (or lower) z-score cut-off.

To point 3 in the previous round of reviews, it is important to include the values of the Spearman correlation coefficients, as requested. The reviewer accepts the a posteriori selection of normalization applied.

REVIEWER COMMENTS

Reviewer #2 (Remarks to the Author):

The authors have addressed all points raised. Thank you.

The authors thank R2 for their insights and support in the publication process.

Reviewer #3 (Remarks to the Author):

The authors have provided detailed replies in attempt to address most of the raised points in the last round of reviews.

Two issues remain:

To point 2 in the previous round of reviews, the authors do not show how robust the findings (from enrichment analyses) are to changing the z-score cut-off. This can be simply done by checking which of the "more interesting" model features remain as such after taking a higher (or lower) z-score cut-off.

Thank you for this additional comment. Robustness is an important concern here. Originally, we varied the threshold from 4% to 8%. Likely due to the large number of models, we saw negligible change in the enrichment statistics and no change in any of the results. Nevertheless, we agree, this is important to mention. We have added a sentence to the methods indicating this check.

To point 3 in the previous round of reviews, it is important to include the values of the Spearman correlation coefficients, as requested. The reviewer accepts the a posteriori selection of normalization applied.

Thank you for reiterating this point. Because correlation and p-value given n (48 samples in cohort 1 and 10 samples in cohort 2) are perfectly defined (see links below), correlation is one of the few statistics that does not need require both effect size and p-value.

That said, and as you point out, requires clarification. To avoid confusion for similarly conscientious readers, we have added correlation values to the figure.

- https://en.wikipedia.org/wiki/Spearman%27s_rank_correlation_coefficient#Determining_significance
- <https://medium.com/analytics-vidhya/spearman-s-correlation-analysis-for-paired-data-b8302f1f4a35>